# THE LAZY NEURON PHENOMENON: ON EMERGENCE OF ACTIVATION SPARSITY IN TRANSFORMERS

**Zonglin Li**[∗]**, Chong You,**[∗] **Srinadh Bhojanapalli, Daliang Li, Ankit Singh Rawat, Sashank J. Reddi, Ke Ye, Felix Chern, Felix Yu, Ruiqi Guo, and Sanjiv Kumar**
Google Research, New York City, USA
`{lizonglin,cyou,bsrinadh,daliangli,ankitsrawat}@google.com`
`{sashank,kkye,fchern,felixyu,guorq,sanjivk}@google.com`

## ABSTRACT

This paper studies a curious phenomenon that machine learning model with Transformer architectures have *sparse* activation maps. By activation map we refer to the intermediate output of the multi-layer perceptrons (MLPs) after a ReLU activation function, and by "sparse" we mean that on average very few entries (e.g., 3.0% for T5-Base and 6.3% for ViT-B16) are nonzero for each input to MLP. Moreover, larger Transformers with more layers and wider MLP hidden dimensions are sparser as measured by the percentage of nonzero entries. Through extensive experiments we demonstrate that the emergence of sparsity is a prevalent phenomenon that occurs for both natural language processing and vision tasks, on both training and evaluation data, for Transformers of various configurations, at layers of all depth levels. We discuss how sparsity immediately implies a way to significantly reduce the FLOP count and improve efficiency for Transformers. Moreover, we demonstrate perhaps surprisingly that enforcing an even sparser activation via Top-$k$ thresholding with a small $k$ brings a collection of desired properties, namely less sensitivity to noisy training data, more robustness to input corruptions, and better calibration for their prediction confidence.

## 1 INTRODUCTION

The great success of modern machine learning for tasks in computer vision, natural language processing, game playing and beyond is driven primarily by the computational model known as deep neural networks (DNNs) (LeCun et al., 2015). With inspirations drawn from biological intelligent systems, DNNs are massive systems of distributed computational nodes (a.k.a. neurons) with learned inter-connections, which possess the capacity of accomplishing complex real-world tasks.

Although originally motivated from biological brains, there are differences at very fundamental levels on how DNNs work compared to biological neural networks. One of such differences is in the sparsity of neural activities. Evidence from neuroscience suggests that neural activity in biological brains is *sparse*, namely, only a small percentage of all neurons fire at each time (Ahmed et al., 2020; Barth & Poulet, 2012; Kerr et al., 2005; Poo & Isaacson, 2009). Sparse firing suggests that despite having billions of neurons, only a small fraction of the brain participates in computation at each time, which may explain why brains can sustain at a very low energy cost. In contrast, learning and inference with DNNs rely primarily on dense computations where all neurons are involved for any input. In fact, modern computational hardware for deep neural networks, such as GPUs and TPUs, are designed to facilitate massive scale dense computations. Even with such dedicated hardware, DNNs are still notoriously resource-demanding to train and deploy. Aside from computation efficiency, DNNs also lag far behind biological brains in terms of robustness to input perturbation, error correction for erroneous training labels, confidence calibration for the predictions, etc.

### 1.1 AN INTRIGUING OBSERVATION: ACTIVATIONS ARE SPARSE IN TRAINED TRANSFORMERS

This paper provides an extensive study on a surprising observation that despite performing dense computations, DNNs produce very *sparse* activation in its intermediate layers once trained. Specifically,

---
∗Equal contribution

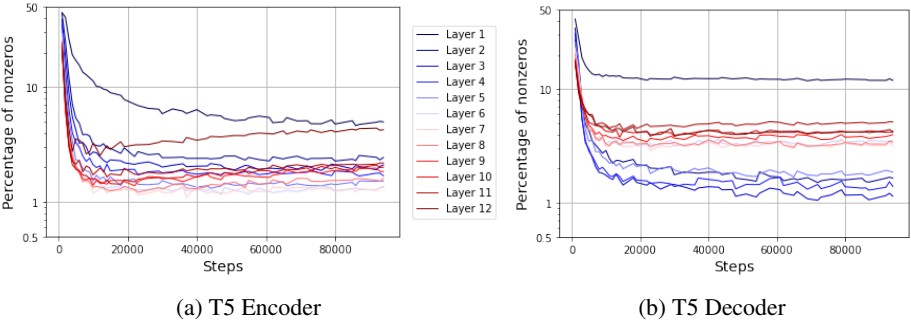

(a) T5 Encoder          (b) T5 Decoder

Figure 1: Percentage of nonzero entries (y-axis, log scale) in the activation map as a function of number of training steps (x-axis) for a T5-Base model trained with the span corruption objective on the C4 dataset. **Left:** layers (from shallow to deep) of the encoder. **Right:** layers of the decoder.

we study *Transformer* (Vaswani et al., 2017), a DNN model architecture that has become a workhorse for modern applications. Transformers are constructed by interleaving a self-attention module and a multi-layer perceptrons (MLPs) of depth 2, and the focus of this paper is on the activation map of the *first* MLP layer. Figure 1 shows the sparsity of the activation maps, measured by the percentage of nonzeros, in all MLP layers of a T5-Base model (Raffel et al., 2020) computed on the training set of C4. We see that the percentage of nonzero entries is around 50% at initialization, which is expected: randomly initialized weights produce roughly equal numbers of positive and negative entries in the pre-activation map, resulting in about 50 % non-zeros after the ReLU. However, at the end of training the percentage of nonzero entries reduces drastically: the average value across all encoder-decoder layers is 2.7% with the largest one being 12.0% and the smallest one being only 1.1%. The emergence of sparse activation in Transformers bears a similarity to the sparsity of neural activities in biological brains, revealing an interesting connection between artificial and biological networks. Moreover, unlike classical sparse methods where such a connection is established via *explicit* sparse regularization (Olshausen & Field, 1996), the sparsity observed in Transformers is emergent without any explicit design.

## 1.2   PREVALENCE, BENEFITS, AND CAUSES OF SPARSITY

This paper studies the aforementioned phenomenon of sparse activation in trained Transformers, with a focus on the following two questions. First, is the phenomenon shown in Figure 1 a corner case or does it occur broadly? Second, why should we care about the sparsity in DNNs, other than the appeal of its similarity to biological brains? Our main results along these two lines are summarized below.

1. **Sparsity is a prevalent phenomenon.** We show in Section 2 that the emergence of sparse activation reported in Figure 1 is not an isolated and cherry-picked case. Rather, sparsity is prevalent, and occurs broadly in Transformer models: it emerges in all layers of a Transformer, for Transformers trained on both vision and natural language data, for Transformers of various configurations, and for activation maps computed on both train and test data, etc. Moreover, through controlled experiments on the width and depth of Transformers, we reveal that larger models are sparser, as measured by percentage of nonzero entries. We also show in the Appendix B that sparsity emerges with many other architectures and with different optimizers.

2. **Sparsity improves efficiency.** Sparsity of activation map in trained Transformers implies that a large proportion of the computation during inference is spent on multiplying values by zero. Hence, FLOPs can be drastically reduced by avoiding all such computations, which we discuss in Section 3.1. Motivated by this observation, and to obtain reduced FLOPs not only after training but throughout training, we introduce *Top-k Transformer* in Section 3.2, a simple modification of Transformers where a Top-$k$ thresholding is applied to the activation maps[1]. We show that Top-$k$ Transformers with a reasonable sized $k$ has on par performance with vanilla Transformers. To demonstrate the computation benefits of Top-$k$ Transformers, we provide proof-of-concept results on wall time reduction for the task of unbatched decoding on TPUv4 with a large Top-$k$ T5. Meanwhile, we emphasise that this result is far from fully realizing the benefit of sparse activation, due to a lack of hardware support for sparse computation.

---

[1]The approach is previously adopted in ConvNets for improving model robustness (Ahmad & Scheinkman, 2019), and more recently in Gupta et al. (2021) for improving memory efficiency of Transformers.

3. **Sparsity improves robustness and calibration.** We further show in Section 3.3 that enforcing explicit sparsity via Top-$k$ Transformers improves model performance in terms of less sensitivity to noisy training data, less sensitivity to input corruptions, and better confidence calibration.

In addition, we provide a study on the causes of sparsity in the Appendix D, showing that sparsity is likely not an artifact of the training data, and may be attributed to the training dynamics in the optimization process.

## 1.3 EXPERIMENTAL SETUP

We study the sparsity in activation maps of Transformers with two commonly used Transformer models, namely Text-to-Text Transfer Transformer (i.e., T5) and Vision Transformer (i.e., ViT).

- **T5** is an encoder-decoder model for natural language processing tasks (Raffel et al., 2020). We train T5 on the Colossal Clean Crawled Corpus (C4) using the span corruption task.
- **ViT** is an encoder model for vision tasks (Dosovitskiy et al., 2021). Unless specified otherwise, we train ViT on ImageNet-21k (Deng et al., 2009), an image classification dataset with 14M images and 21k classes. For certain cases we also use ImageNet-1k which is a subset of ImageNet-21k with 1.3M images and 1k classes.

We measure the sparsity level (computed on training set unless specified otherwise) at the intermediate output of the two-layer MLPs in a Transformer . Recall that an MLP performs the following mapping

$$f(\boldsymbol{x}; \boldsymbol{K}, \boldsymbol{V}) \doteq \sum_{i=1}^{d_{\text{ff}}} \Big( \sigma(\langle \boldsymbol{k}_i, \boldsymbol{x} \rangle) \cdot \boldsymbol{v}_i \Big), \text{ or equivalently, } f(\boldsymbol{x}; \boldsymbol{K}, \boldsymbol{V}) \doteq \boldsymbol{V} \sigma(\boldsymbol{K}^\top \boldsymbol{x}), \quad (1)$$

where $\boldsymbol{x} \in I\!\!R^{d_{\text{model}}}$ is the input, $\boldsymbol{K} = [\boldsymbol{k}_1, \ldots, \boldsymbol{k}_{d_{\text{ff}}}] \in I\!\!R^{d_{\text{model}} \times d_{\text{ff}}}$ and $\boldsymbol{V} = [\boldsymbol{v}_1, \ldots, \boldsymbol{v}_{d_{\text{ff}}}] \in I\!\!R^{d_{\text{model}} \times d_{\text{ff}}}$ are learnable layer parameters, and $\sigma()$ is a nonlinear activation function. We use ReLU as the activation function $\sigma()$ for both T5 and ViT[2]. A two-layer MLP may be regarded as having $d_{\text{ff}}$ neurons in the hidden layer, where the $i$-th neuron performs the computation $\sigma(\langle \boldsymbol{k}_i, \boldsymbol{x} \rangle) \cdot \boldsymbol{v}_i$, and the final layer output is the sum of the output of all neurons. Each neuron is called *activated* if $\sigma(\langle \boldsymbol{k}_i, \boldsymbol{x} \rangle)$ is strictly positive. Hence, the sparsity of neuron activation can be measured by the number of nonzero entries in the feature map

$$\boldsymbol{a} \doteq \sigma(\boldsymbol{K}^\top \boldsymbol{x}) \in I\!\!R^{d_{\text{ff}}}. \quad (2)$$

Both T5 and ViT come with several configurations for $d_{\text{model}}, d_{\text{ff}}$, number of layers, etc. Unless specified otherwise, we will use the Base models (i.e., T5-Base and ViT-B/16) which have $d_{\text{model}} = 768$, $d_{\text{ff}} = 3072$, and 12 layers (for ViT) and 12 encoder layers $+12$ decoder layers (for T5). Our experiment with T5 and ViT uses the T5X (Roberts et al., 2022) and the Scenic codebase (Dehghani et al., 2022), respectively. More training details of T5 and ViT are provided in Appendix A.

## 2 PREVALENCE OF SPARSITY IN LEARNED TRANSFORMERS

This section shows thorough experiments on commonly used Transformers that sparsity in activation maps is a prevalent phenomenon. We also show through some controlled experiments that deeper and wider Transformers tend to be sparser measured by percentage of nonzero entries in activation maps.

## 2.1 SPARSITY IS A UBIQUITOUS PHENOMENON

We start by providing experimental evidence that the emergence of sparse activation in trained Transformers is a ubiquitous phenomenon. To this end, we plot the percentage of nonzero entries of activation maps in different Transformers, and present the results in Figure 2. These results demonstrate the following.

- *Sparsity emerges for both Vision and NLP tasks.* Figure 2a shows the percentage of nonzero entries of trained T5 and ViT models evaluated on their respective training datasets. We see that both encoder and decoder of T5, as well as the ViT, all exhibit sparsity.

---

[2]ViT uses GeLU as its activation function (Dosovitskiy et al., 2021). Here we switch to ReLU as it allows us to more easily measure the sparsity level using the number of nonzero entries with a very small performance drop (e.g., $47.78\%$ with GeLU vs $47.58\%$ with ReLU for Top-1 evaluation accuracy on ImageNet-21K).

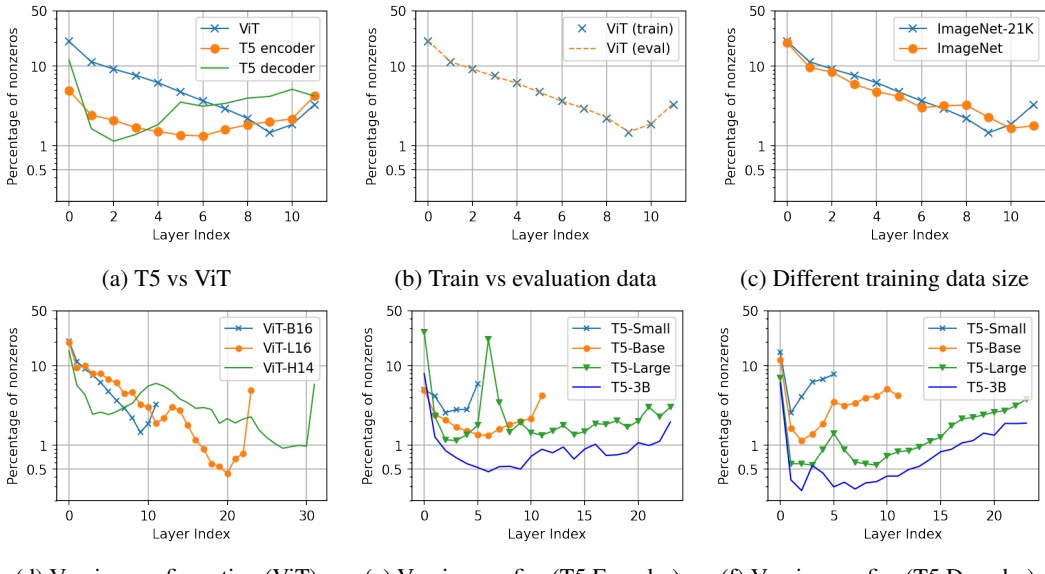

(a) T5 vs ViT  (b) Train vs evaluation data  (c) Different training data size

(d) Varying configuration (ViT)  (e) Varying config. (T5 Encoder)  (f) Varying config. (T5 Decoder)

Figure 2: Percentage of nonzero entries across different layers of trained Transformers (a) for both language data with T5 and vision data with ViT, (b) on both train and evaluation data, (c) for ViT trained on ImageNet of 21k vs 1k classes, (d) on ViT of varying configurations, and (e, f) on T5 of varying configurations. Note that the y-axis is in $\log$ scale. *Sparsity emerges in all cases.*

- *Sparsity emerges on both training and evaluation data.* Figure 2b shows the percentage of nonzero entries in a trained T5 model measured on both the training data and the evaluation data. We see that the property of sparsity generalizes very well to evaluation data as the curves for training and evaluation data align very closely with each other.

- *Sparsity emerges on datasets of varying scale.* Figure 2c shows the percentage of nonzero entries in ViT trained on both ImageNet-21k and ImageNet-1k, where the former is a superset of the later with approximately $10\times$ more images and $21\times$ more classes. We see that the scale of data does not affect much of the sparsity level.

- *Sparsity emerges on Transformers of varying configurations.* Figure 2d shows the percentage of nonzero entries for ViT of varying configurations in model size. Figure 2e and 2f show the percentage of nonzero entries for encoder and decoder, respectively, of T5 with varying configurations in model size. We see that sparsity persists for all cases.

- *Sparsity emerges across all layers of a Transformer.* Finally, all plots in Figure 2 show that sparsity emerges in all layers of a Transformer. Moreover, in all cases the first few and last few layers tend to be denser than intermediate layers.

The presence of sparsity in activation maps does not rule out the possibility that a small percentage of the neurons are always activated for all inputs, whereas the rest of the neurons are never activated. To illustrate that this is not the case, we experiment with a pretrained T5 base model[3] to plot the percentage of layer inputs for which each of the $d_{ff}$ neurons is activated when evaluated on 800 examples taken from C4 dataset with span corruption task. Note that there are $800 \times 512 = 409600$ samples as MLP activation is computed per token. The results are presented in Figure 3 with x-axis being indices of neurons in the first encoder layer of T5 sorted in descending order according to percentage of layer inputs on which they are activated. It can be seen that while a few neurons are activated for around 50% of the time, the vast majority of neurons (around 93.5%) are activated less than 10% of the time. Moreover,

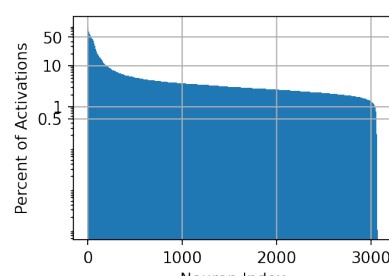

Figure 3: Percentage of times that each neuron in the first MLP layer of a trained T5 is activated on C4 dataset.

---

[3] https://github.com/google-research/t5x/blob/main/docs/models.md#t5-checkpoints

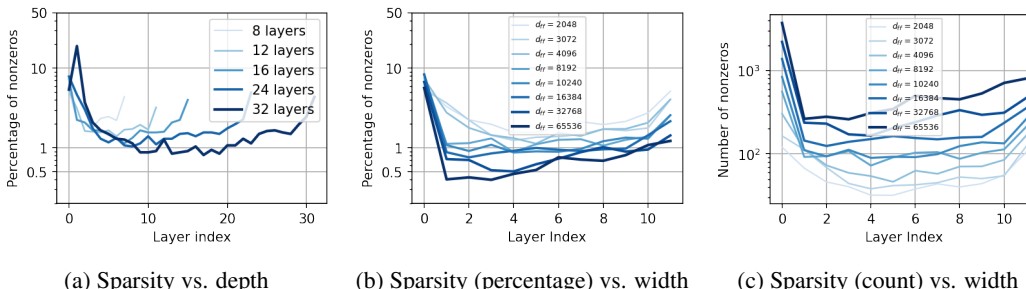

| (a) Sparsity vs. depth | (b) Sparsity (percentage) vs. width | (c) Sparsity (count) vs. width |

Figure 4: Activation sparsity across different encoder layers of trained T5 Transformers of (a) varying depth and (b, c) varying width (i.e., $d_{ff}$). Since with varying width the dimension of activation maps also changes, we evaluate sparsity both in term of the percentage (as in (b)) and the count (as in (c)) of nonzeros. *Deeper and wider models are sparser in terms of percentage of activated neurons.*

there are no dead neurons that are never activated, and the least activated neuron is activated for around 0.001% of the time, and 99% of neurons are activated over 1% of the time. Finally, while the results here are for neurons in the first MLP layer of a pretrained T5 base encoder, all other MLP layers show qualitatively similar behavior.

## 2.2 THE LARGER, THE SPARSER

We next examine the effect of model size on the sparsity level of activation maps. Note that Figure 2e and Figure 2f provide evidence with T5 of varying configuration that larger models tend to be sparser. Here we perform controlled experiments to examine the effect of model depth, measured by the number of Transformer layers, as well as the effect of model width, measured by the dimension of activation map of MLPs (i.e., $d_{ff}$), separately. Towards that, we take a standard T5 model and vary the depth and width, respectively while keeping the rest of the configuration fixed, and examine their sparsity level after training. The results are presented in Figure 4 for the encoder, whereas we omit the results for the decoder as they are qualitatively the same as those for encoder.

It can be seen from Figure 4a that deeper Transformers are arguably sparser. For example, many of the middle layers of the 32-layer model have less than 1% nonzero entries while all shallower models have more than 1% nonzero entries across all layers. For comparing networks of different widths, we measure the sparsity with the percentage and the count of nonzero entries in Figure 4b and Figure 4c, respectively. It can be seen that wider models have a lower percentage of nonzero entries, though a higher count of nonzero entries.

## 3 EFFICIENT, ROBUST, AND CALIBRATED: SPARSITY IS ALL YOU NEED?

In this section we show that activation sparsity provides several practical benefits. In Section 3.1 we discuss how the free sparsity in trained Transformers brings us free computation efficiency in terms of FLOPs count *during inference*. In order to obtain sparsity hence FLOPs reduction *throughout training*, in Section 3.2 we introduce Top-$k$ Transformers, a simple modification of Transformers where a top-$k$ thresholding operation is applied to the activation maps in all MLPs. While existing hardware cannot well support sparse computation and fully realize the benefit of FLOPs reduction, we provide a proof-of-concept experiment on preliminary benefits of Top-$k$ Transformer. Finally, in Section 3.3 we show that sparsity in activation is a good regularization for Transformers. Namely, enforcing sparser activation with smaller values of $k$ in Top-$k$ Transformer (without any other hacks, tweaks and hyperparameter tuning) bestows Transformers several desired properties, namely, robustness of training with erroneous annotations, less sensitivity to input noise/perturbation, and better confidence calibration of the predictions.

## 3.1 EFFICIENCY FOR FREE

Given an embedding dimension $d_{model}$ and an MLP intermediate dimension $d_{ff}$, the computational complexity of a Transformer for an input sequence of length $N$ is $\mathcal{O}(Nd_{model}^2 + N^2 d_{model} + N d_{model} d_{ff})$, where the first term comes from computing the key, query, and value matrices, the second term comes from computing the self-attention matrix, and the third term comes from the MLP. For a fixed

sequence length $N$, and considering the fact that $d_{\text{ff}}$ is often much larger than $d_{\text{model}}$, it is arguable that MLP poses the computational bottleneck in large Transformers. In the following, we explain how sparsity in activation map of MLP can be leveraged to significantly reduce its computational cost, without affecting the model performance.

**Efficiency for the Second MLP Layer.** The sparse activation immediately suggests that a lot of the computation for inference with Transformers is not needed at all. That is, while doing dense matrix-matrix multiplications, much of it is about multiplying a vector by a value of zero, which can be avoided to save computation.

Specifically, we consider the second layer of the MLP in (1) which performs the computation

$$\boldsymbol{V}\boldsymbol{a}, \tag{3}$$

where $\boldsymbol{a} \in \mathbb{R}^{d_{\text{ff}}}$ is the intermediate activation map of MLP (see (2)) and $\boldsymbol{V} \in \mathbb{R}^{d_{\text{model}} \times d_{\text{ff}}}$ is the layer parameter. Eq. (3) involves a simple matrix-vector multiplication which has a FLOP count of $2d_{\text{model}} \times d_{\text{ff}}$. However, if $\boldsymbol{a}$ is sparse with, say $s$ nonzero entries, then the FLOP count for (3) reduces to $2d_{\text{model}} \times s$. Hence,

FLOP in the second MLP layer is reduced by a factor of $1 - \frac{s}{d_{\text{ff}}}$.

Note that $\frac{s}{d_{\text{ff}}}$ is exactly the percentage of nonzeros plotted in the y-axis of e.g. Figure 1, which is $2.7\%$ averaged across all layers. Hence, the computational cost of the second MLP layer can be reduced by a significant amount. More excitingly, the reduction factor $1 - \frac{s}{d_{\text{ff}}}$ is likely to be even bigger for larger Transformer models (see Figures 4a and 4b), pointing to a greater reduction in computation.

**Efficiency for the First MLP Layer.** The sparsity in the intermediate activation map of MLP does not immediately suggest a reduction in computation for the first MLP layer. Nonetheless, it is possible to significantly reduce the computation in the first MLP layer by leveraging approximate nearest neighbor search, which we explain next.

Recall from (1) that the computation in the first MLP layer is given by

$$\sigma(\boldsymbol{K}^{\top}\boldsymbol{x}), \tag{4}$$

with $\boldsymbol{K} = [\boldsymbol{k}_1, \ldots, \boldsymbol{k}_{d_{\text{ff}}}] \in \mathbb{R}^{d_{\text{model}} \times d_{\text{ff}}}$ being the layer parameter and $\boldsymbol{x}$ being the layer input. If the output is sparse with $k$ nonzero entries, then the calculation in (4) may be formulated as finding $k$ points from the set $\{\boldsymbol{k}_i\}_{i=1}^{d_{\text{ff}}}$ that are "closest" to the input $\boldsymbol{x}$ measured by values of inner product. Such a problem is well-known as the nearest neighbor search (NNS) problem or the maximum inner product search problem. While naive solution of the NNS problem has linear complexity in $d_{\text{ff}}$, there exists *approximate* algorithms (Guo et al., 2020; Johnson et al., 2019; Shrivastava & Li, 2014) that are of sublinear complexity, and using them in Transformers means that

FLOP in the first MLP layer may be reduced to have sublinear complexity in $d_{\text{ff}}$.

There are of course the questions of whether such approximate NNS algorithms could hurt Transformer performance, which we leave for future study.

## 3.2 SPARSITY IN TRAINING VIA TOP-$k$ TRANSFORMERS

The benefit of efficiency from sparsity in Section 3.1 comes with caveats. First, while the activation maps are sparse on average, there is the possibility that some of the activation maps for certain inputs are denser hence cannot benefit from sparse computation. Second, sparsity occurs only in trained Transformers while the computation is dense during and particularly at the beginning of training.

Here we present Top-$k$ Transformer, a simple modification to Transformer architecture that allows us to control sparsity level for all model inputs, and throughout training. Top-$k$ Transformer is built upon a regular Transformer with the only modification being the MLP layers, where at the output of the activation function $\sigma()$ (see (1)) we add a Top-$k$ thresholding operator. That is, the MLPs of Top-$k$ Transformers perform the following computation

$$f(\boldsymbol{x}; \boldsymbol{K}, \boldsymbol{V}) = \boldsymbol{V} \cdot \text{Top}_k\Big(\sigma(\boldsymbol{K}^T\boldsymbol{x})\Big), \tag{5}$$

where $\text{Top}_k(\cdot)$ performs a thresholding that all entries other than those of the largest $k$ values are set to zero with $k$ being a hyper-parameter subject to design choices. Note that Top-$k$ Transformer reduces to a regular Transformer if we set $k = d_{\text{ff}}$. By using a small value of $k$, the benefit of efficiency in terms of reduction in FLOP as discussed in Section 3.1 applies to Transformer training as well.

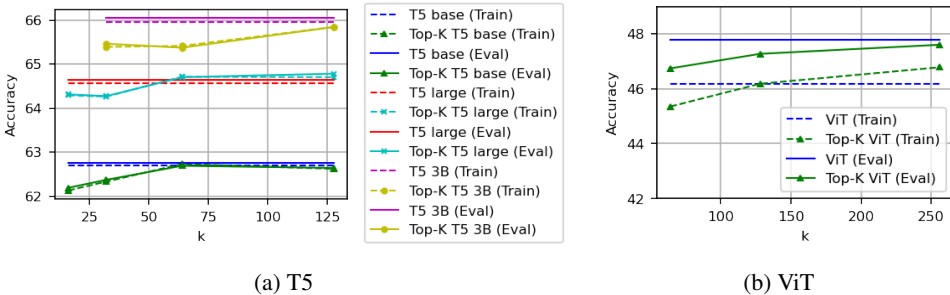

(a) T5                                          (b) ViT

Figure 5: Training and evaluation accuracy of Top-$k$ T5 for three different sizes: base, large and 3B (left) and Top-$k$ ViT (right) with varying $k$. *Top-$k$ Transformer is on par with regular Transformer for a large enough $k$*. e.g. for T5 3B with $k = 128$, and ViT with $k = 256$, the drop is around $0.3\%$.

The immediate question for Top-$k$ Transformer is whether it offers training sparsity at the cost of a reduced performance. Here we conduct experiments with Top-$k$ T5 and Top-$k$ ViT, and evaluate their performance measured by prediction accuracy for C4 span corruption and ImageNet-21k classification tasks, respectively. The results are provided in Figure 5. We see that with the Top-$k$ T5-{Base, Large, 3B} (resp., Top-$k$ ViT) Transformer, taking $k$ to be 128 (resp., 256) is sufficient for closely matching the test performance of the vanilla T5-{Base, Large, 3B} (resp., ViT). Note that this is achieved without any other hyper-parameter tuning for the Top-$k$ Transformers upon those used for a regular Transformer, and other hyper-parameter choices may further improve the performance of Top-$k$ Transformers.

We now provide experimental results with Top-$k$ Transformers on wall-time benefits from FLOPs reduction discussed in Section 3.1. In particular, we evaluate the inference time latency reduction of Top-$k$ Transformer. In our experiment, we add a Top-$k$ thresholding to T5X (Roberts et al., 2022)[4]. We gain efficiency in the second MLP layer by an implementation that avoids all multiplication by zero as described in Section 3.1. The decoder per-token wall time for unbatched greedy decoding during inference on a single TPUv4 chip is presented in Figure 6. We observe that larger models have more wall time reduction, due to the fact that they have larger $d_{\text{ff}}$ hence more FLOPs reduction. In particular, for T5-11B we observe around $10\%$ wall time reduction with $k \leq 128$, though this amount becomes smaller with a larger $k = 256$.

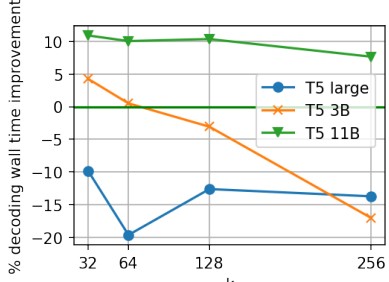

Figure 6: Latency reduction for unbatched greedy decoding in decoder of Top-$k$ Transformers on TPUv4.

Finally, we emphasize that the sparsity in Top-$k$ Transformers is *unstructured* and data-dependent, which is not well supported on existing computation hardwares such as TPUs and GPUs. Hence, the results in Figure 6 are for proof-of-concept purposes, and are far from fully realizing the benefit of FLOPs reduction via sparsity. We leave a study of better implementation of sparse computation for obtaining higher wall time reduction to future work.

## 3.3    BONUS! IMPROVED ROBUSTNESS AND CALIBRATION

Despite not being explicitly designed for such purposes, inducing sparse activation via Top-$k$ Transformer has the benefits of improving model robustness[5] and confidence calibration. We demonstrate this using the image classification task with the ImageNet-1k dataset, and present the results in Table 1. All results for Top-$k$ ViT are obtained without any model and training hyper-parameter tuning upon those for ViT. Contexts and details are presented below. More results are presented in Appendix C.

---

[4]We use the implementation of `jax.lax.approx_max_k` (Chern et al., 2022) with a recall target of 0.95.
[5]This is previously demonstrated in Ahmad & Scheinkman (2019) for ConvNets.

Table 1: Evaluation of Top-128 ViT for ImageNet-1k classification in terms of 1) natural accuracy with ImageNet-1k evaluation set, 2) robust accuracy with {40%, 80%} corrupted training labels, 3) robust accuracy under input perturbation with additive {Gaussian, Impulse, Shot} noise on evaluation images, and 4) calibration error on evaluation data measured by ECE. *Top-128 ViT is on par with ViT for natural accuracy while is significantly better for model robustness and calibration.*

| Methods | Natural Accuracy | Accuracy w/ Train Label Noise | | Accuracy under Input Perturbation | | | Expected Calibration Error (ECE) |
|---------|------------------|-------|-------|----------|---------|-------|------------------|
| | | 40% | 80% | Gaussian | Impulse | Shot | |
| ViT | 74.85% | 59.44% | 25.35% | 39.54% | 37.37% | 38.56% | 8.42% |
| Top-128 ViT | 74.83% | 62.13% | 30.80% | 42.29% | 40.07% | 40.68% | 7.48% |

**Robustness to Label Noise.** An important challenge for DNNs is that they are highly susceptible to label noise, the problem where a certain percentage of training labels are corrupted or erroneously generated. This may be attributed to the fact that DNNs are often over-parameterized, hence too "capable" that they tend to overfit, or "memorize" the noisy labels without generalizing to test data. While many dedicated techniques exist (see e.g., Algan & Ulusoy (2021); Song et al. (2022) for a review), here we show that a simple Top-$k$ Transformer can effectively address the label noise issue.

We conduct experiments using the ImageNet-1k dataset for which we replace $p\%$ of the labels in the training set with a random label drawn uniformly from the set of all possible labels. The evaluation performance under $p \in \{40\%, 80\%\}$ label noise is presented in Table 1. It shows that Top-$k$ offers a consistent performance gain with label noise.

**Confidence Calibration.** Aside from label noise, another symptom of over-parameterization of DNNs is that they tend to be overly confident in their predictions. In the context of classification problems, they tend to assign a high (i.e., close to 1) probability to the class of its prediction, while it is more desirable that they produce a probability that is commensurate with its confidence level (Guo et al., 2017). A commonly used metric for confidence calibration is the expected calibration error (ECE) (Naeini et al., 2015), which is the discrepancy between the probability to the class of a model's prediction and the probability that its prediction is actually correct.

Here we measure the calibration of Top-$k$ ViT via ECE and report the results in Table 1. It shows that Top-$k$ with $k = 128$ enables the Transformer to be more calibrated when compared to a vanilla Transformer. Furthermore, results reported in Appendix C show that ECE monotonically decreases as $k$ is decreased from 128 to 32.

**Robustness to Input Perturbation.** Another important challenge with DNNs is that their outputs tend to be sensitive to naturally occurring image corruptions, which limits their application to mission critical tasks (Bhojanapalli et al., 2021). Here we evaluate the robustness of Top-$k$ ViT to three types of additive noises, namely Gaussian noise, impulse noise, and shot noise. For that purpose, we train Top-$k$ ViT on standard ImageNet-1k training data and report their classification accuracy on ImageNet-C (Hendrycks & Dietterich, 2019), a benchmark that contains algorithmically generated Gaussian, impulse, and shot noise (among many others types) applied to the ImageNet-1k test dataset. For each noise type, there are five severity levels. We report the averaged performance over all severity levels of each corruption type in Table 1 for $k = 128$, and in Appendix C for a few other values of $k$. We see that robust accuracy is the highest with $k = 64$, while taking $k = 128$ or $k = 32$ also provides benefits compared to the vanilla Transformer.

## 4 RELATED WORK

Prior efforts on introducing sparsity in deep neural networks abound, though often with diverse motivations and objectives. Here we provide a brief overview of several popular lines of work.

**Sparsity for Efficiency.** Sparsity in either model weights or activation maps is often used for improving training and inference efficiency (see e.g. Hoefler et al. (2021) for a review). For activation sparsity in particular, sparsity for efficiency is explored perhaps first in ConvNets (Georgiadis, 2019; Kurtz et al., 2020; Rhu et al., 2018) before subsequently becoming a key design component in many of the largest Transformer based language and vision models (Du et al., 2022; Fedus et al., 2022a;b; Rajbhandari et al., 2022). The Top-$k$ thresholding that we use in Top-$k$ Transformer has also been

previously used in Gupta et al. (2021) to improve memory efficiency of Transformers. However, it has been unclear *a priori* whether sparsity hurts model performance, hence the practice often relies on wishful design, trial-and-error, and post-hot justification (Baykal et al., 2022). Our discovery that Transformers naturally produce sparse activation maps, and that larger models are even sparser, may provide principled perspectives towards efficiently training future large models.

**Sparsity for Robustness.** Many works find that smaller and sparser networks obtained by model compression are more robust to adversarial perturbation (Chen et al., 2022; Guo et al., 2018; Jordao & Pedrini, 2021) and label noise (Xue et al., 2022). Another line of work that uses sparsity for robustness leverages the property that practical data corruption is often sparse (Ghosh et al., 2017; Liu et al., 2022; You et al., 2020). None of the work mentioned above is based on sparsity in activation maps. More closely related to ours is the work of Ahmad & Scheinkman (2019) where sparsity in activation map of convolutional DNNs is shown to improve robustness to input perturbation, and Muthukumar & Sulam (2022) that leverages sparse activation to derive robust generalization error bounds.

**Sparsity for Explainability.** Work on leveraging sparsity for interpreting deep learning models long exist but often in a post-hoc fashion for examining the semantic meanings encoded by a neuron of a trained model (Dalvi et al., 2019). For Transformers, evidence suggests that the learned knowledge is encoded mainly in its MLPs with individual neurons expressing specific factual knowledge (Dai et al., 2022). Moreover, enforcing neuron activation sparsity in MLPs helps to improve the percentage of neurons that are interpretable (Elhage et al., 2022). Hence, our discovery may point to new directions towards developing more interpretable DNNs (Cuadros et al., 2022; Sajjad et al., 2021).

**Sparsity for Data Modeling.** Following the seminal work of Olshausen & Field (1996), there are a lot of interests in sparsity as an effective modeling of natural signals (Mairal et al., 2014). With the close resemblance of the computational structure of ReLU networks and sparse encoding algorithms (Gregor & LeCun, 2010), it became natural to study a DNN as a multi-layer sparse modeling of the data (Papyan et al., 2018). Along with substantial theoretical understanding of such a modeling are obtained (Papyan et al., 2017; Sulam et al., 2018), there are also experimental results on their practical benefits (Sun et al., 2018) though less often on modern large-scale data.

**Sparsity for Theory of Over-parameterized Models.** Because of its simplicity and well-develped theory in classical machine learning (Candès & Wakin, 2008; Vidal et al., 2015; Wright & Ma, 2022), sparse modeling is often used to provide theoretical understanding of modern large and over-parameterized models. This include works on implicit regularization (Chou et al., 2021; Nacson et al., 2022; Vaskevicius et al., 2019; Woodworth et al., 2020; Zhao et al., 2019), nonconvex optimization (Buhai et al., 2020; Sulam et al., 2022), noise interpolators (Chinot et al., 2022; Donhauser et al., 2022; Koehler et al., 2021), etc. However, the aforementioned work uses sparsity as a testbed or toy model to gain insights, without implication of existence of sparsity in DNNs.

## 5 DISCUSSION

This work demonstrates the natural emergence of sparse activation in commonly used Transformer models (Section 2). The notion of sparsity pertains to the law of parsimony, a.k.a. *Occam's razor*, where among all possible explanations of observed data, the *simplest* ones are preferred. It is a fundamental scientific principle broadly used in various scientific and engineering subjects (Domingos, 1999; Epstein, 1984), including classical machine learning (Tibshirani, 1996). Hence, our discovery may be suggesting that the law of parsimony is playing a role in Transformers even though they are not explicitly designed so, resonating with recent view on the role of sparsity for intelligence systems (LeCun, 2022; Ma et al., 2022; Roberts, 2021; Vasudevan et al., 2021). More importantly, we back such a perspective by providing evidence of improved robustness and calibration via enforcing sparsity using Top-$k$ thresholding (Section 3), which indicates that sparsity is indeed a pertinent prior for good generalization. We hope that our work may motivate future effort on introducing sparsity in deep learning models in a more principled way for obtaining more efficient, robust, and calibrated models. Finally, while our motivation of studying sparse activation in Transformers comes (partly) from study of biological brains, establishing such a connection may reciprocally benefits efforts on applying artificial intelligence to the study of biology and neuroscience (Richards et al., 2022).

## ACKNOWLEDGMENTS

We would like to acknowledge helpful discussions with René Vidal and Jeremias Sulam from Johns Hopkins University, with Weijie Su from UPenn, with Yuxiang Wang from UC Santa Barbara, with Atlas Wang from UT Austin, with Nishanth Dikkala, Nikhil Vyas, Preston McAfee and Mukund Sundararajan from Google, with Subutai Ahmad from Numenta, with Wei Hu, Salar Fattahi, and Jianhao Ma from University of Michigan, with Tuo Zhao from Georgia Tech. We particularly thank Donhauser Konstantin from ETH Zurich for interesting discussion on hypothesis for emergence of sparsity.

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

# Appendices

The appendices are organized as follows. In Section A we provide the implementation details for experiments conducted in this paper. In Section B we demonstrate the emergence of sparse activation in other architectures and with other optimizers than those used in Section 2. In Section C we provide additional experiments upon those in Section 3 to demonstrate the benefits of sparsity. In Section D we explore the potential causes of sparsity, with a focus on the effect of training data. In Section E we present a derivation to show that during early training, the final MLP layer's intermediate activation tends to get sparse. Finally in Section F we present insights on the emergence of activation sparsity from experiments on 2-layer MLP models.

## A    IMPLEMENTATION DETAILS

### A.1    T5

For most of the experiments, except the Top-$k$ transformer, we used vanilla T5 architecture (Raffel et al., 2020). We trained model with Adafactor optimizer, an inverse square root learning rate schedule, and no dropout. For the first 10,000 steps we also use a fixed learning rate of 0.01 as warm-up. The training task is span corruption without any mixture, and unless specified otherwise, we train the model for 100,000 steps with batch size of 256 to save compute and time, as the sparsity or accuracy trend is already clear by then. We used 512 tokens on the encoder side and 114 tokens on the decoder side.

### A.2    VIT

Following Dosovitskiy et al. (2021), we train ViT using ADAM (Kingma & Ba, 2015) as the optimizer with $\beta_1 = 0.9, \beta_2 = 0.999$. Other training details such as weight decay, dropout rate, and learning rate all follow the description in (Dosovitskiy et al., 2021, Section B.1) except that we train for 180 epochs (as opposed to 300) on ImageNet-1k.

### A.3    T5 / VIT CONFIGURATIONS

For the reader's convenience, we summarize the configuration of varying T5 / ViT models used in our paper in Table A.1.

Table A.1: Configuration of T5 and ViT that are used in the experiments. $d_{\text{model}}$ and $d_{\text{ff}}$ are defined in Section 1.3. # Layers is the number of encoder + decoder layers for T5 and encoder layers for ViT.

|  | | T5 | | | | | ViT | |
|---|---|---|---|---|---|---|---|---|
|  | Small | Base | Large | 3B | 11B | Base | Large | Huge |
| $d_{\text{model}}$ | 512 | 768 | 1024 | 1024 | 1024 | 768 | 1024 | 1280 |
| $d_{\text{ff}}$ | 2048 | 3072 | 4096 | 16384 | 65536 | 3072 | 4096 | 5120 |
| # Layers | 6 + 6 | 12 + 12 | 24 + 24 | 24 + 24 | 24 + 24 | 12 | 24 | 32 |
| # Parameters | 60M | 220M | 770M | 2,800M | 11,000M | 86M | 307M | 632M |

## B    ADDITIONAL RESULTS ON PREVALENCE OF SPARSITY

### B.1    SPARSITY AND NETWORK ARCHITECTURE

We evaluate the sparsity level of activation map in several commonly used network architectures beyond T5 and ViT. This includes BERT which is also a Transformer based architecture, as well as non-Transformer based architectures such as MLP-Mixer and ConvNets. We also examine whether residual connection accounts for the emergence of sparsity.

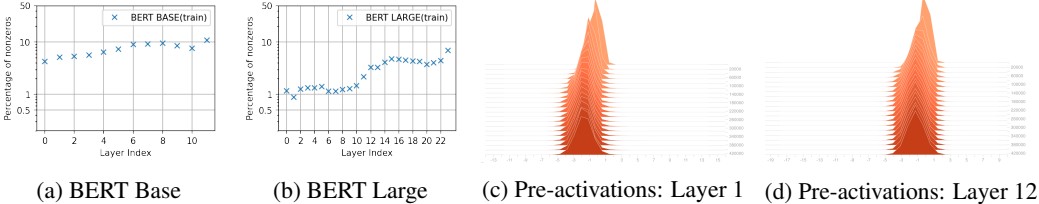

| (a) BERT Base | (b) BERT Large | (c) Pre-activations: Layer 1 | (d) Pre-activations: Layer 12 |

Figure B.2: Plots a, b: Percentage of nonzero entries in activation maps of BERT Base and Large models (Devlin et al., 2019) trained on Wikipedia dataset. We observe high levels of sparsity (<10%) similar to other Transformer models. Plots c, d: Histograms of pre-activation values for layers 1 and 12 of a Bert Base model. We notice that while at initialization the activations are distributed with mean 0, the mean quickly shifts negative as the training progresses, resulting in high levels of sparse activation values.



| (a) GeLU Activation | (b) Sigmoid Activation | (c) Tanh Activation |

Figure B.3: Layer 1 Preactivation histograms for BERT Base models with different activation functions. We observe similar behavior as ReLU with GeLU and Sigmoid activations. However Tanh activation has different distribution of preactivation values. The network doesn't show sparsity and the accuracy is also worse in comparison to ReLU/GeLU.

**BERT.** We evaluate the sparsity level of BERT models (Devlin et al., 2019). We specifically consider BERT Base (12 layers) and BERT Large (24 layers) Transformer models, with ReLU activation in the MLP layers. We follow the same training receipe as Devlin et al. (2019) and pre-train these models on Wikipedia and Books dataset using Masked Language Modelling (MLM) objective. We train for 450000 steps with a batch size of 1024 using AdamW optimizer with $1e-4$ learning rate.

In Figure B.2 we plot the sparsity levels of both BERT models for all the intermediate MLP layers (plots a and b). We observe that both these models exhibit high levels of sparsity ($< 10\%$) as other Transformer models. We further visualize the pre-activation values of the MLP layers as histograms in plots c and d. We observe that while they have mean 0 at initialization, the mean quickly becomes negative as training progresses, resulting

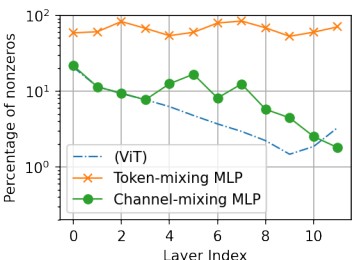

Figure B.1: Percentage of nonzero entries in activation maps of MLP-Mixer trained on ImageNet-21k. Results for token-mixing and channel-mixing MLPs are plotted in separate curves.

in high sparsity levels. Finally, in Figure B.3 we provide a visualization of pre-activation of values with several popular activation functions, including GeLU, Sigmoid, and Tanh. We observe a similar distribution of preactivation values as ReLU with GeLU and Sigmoid activations. However Tanh activation has a different distribution of preactivation values. With Tanh activation, the network does not show sparsity and the accuracy is significantly worse in comparison to ReLU/GeLU.

**MLP-Mixer.** We evaluate the sparsity level of the MLP-Mixer (Tolstikhin et al., 2021), an all-MLP architecture constructed from cascading token-mixing and channel-mixing MLPs. Specifically, we use Mixer-B16 as the architecture, ADAM with $\beta_1 = 0.9, \beta_2 = 0.999$ as the optimizer, and train on ImageNet-21k for 300 epochs. While Tolstikhin et al. (2021) sweeps over a product set of hyper-parameters, here for simplicity we use a fixed set of hyper-parameters with weight decay of 0.03, gradient norm clipping at 1.0, base learning rate of 0.003, RandAugment magnitude of 10, no mixup, no stochastic depth, and no dropout.

Figure B.1 shows the sparsity level at the intermediate layer of both token mixing and channel mixing MLPs of Mixer-B16. We also plot the sparsity level of ViT (i.e., the plot in Figure 2a) to Figure B.1

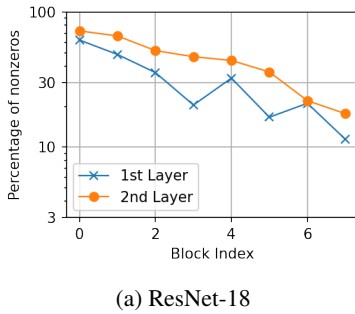
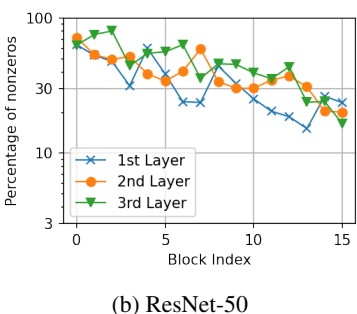

(a) ResNet-18             (b) ResNet-50

Figure B.4: Percentage of nonzero entries in activation maps of ResNet-18 and ResNet-50 trained on ImageNet-1k. Results for the two (resp., three) layers in each residual block (resp., bottleneck residual block) of ResNet-18 (resp., ResNet-50) are plotted in separate curves.

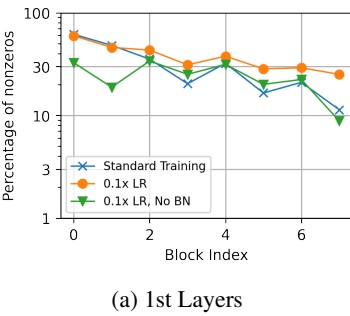
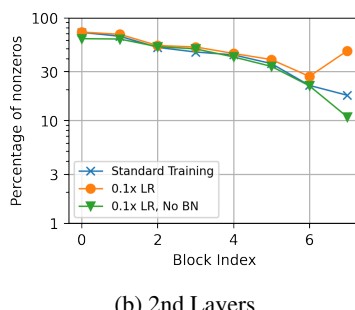

(a) 1st Layers             (b) 2nd Layers

Figure B.5: Effect of batch normalization (BN) on sparsity level across layers of ResNet-18. Because ResNets cannot be effectively train without BN, we reduce the learning rate (LR) by a factor of 10, and multiply the residual branch by a trainable scalar initialized at 0.

for a comparison. It can be seen that the first four layers of channel mixing MLP and ViT have almost identical sparsity levels, while the rest of the layers (other than the last one) of channel mixing MLP are denser than the corresponding layer of ViT. On the other hand, the token mixing MLPs produce dense activation maps with more than 50% nonzero entries, probably because the dimension of the activation maps (384) is too small.

**Convolutional Neural Network (ConvNet).** Sparsity in activation maps has been studied for ConvNets such as the AlexNet (Krizhevsky et al., 2017) at least as early as in the work of Rhu et al. (2018). There are also follow-up work (Georgiadis, 2019; Kurtz et al., 2020) on how enforcing sparse activation maps can help to gain computation efficiency. For completeness, we evaluate and present results for the sparsity level of residual networks (ResNets) (He et al., 2016), which is one of the most commonly used ConvNets, trained on ImageNet-1k. In particular, we focus on ResNet-18 and ResNet-50 which are constructed from stacking 8 standard residual blocks and 16 "bottleneck" residual blocks, respectively, where each block has two and three convolutional and ReLU layers, respectively. We examine the sparsity of activation maps after each of the ReLU layers in each residual block.

The results for ResNet-18 and ResNet-50 are reported in Figure B.4a and Figure B.4b, respectively. Here, the x-axis is the index of the residual block, and the sparsity of different layers in the residual blocks are plotted with separated curves in each figure. It can be observed that

- Layers near the network output tend to produce sparser activation maps than layers near the network input. This is aligned with the observation with ViT trained on ImageNet-1k (see Figure 2b).
- For each residual block, the intermediate layers (i.e., the 1st layer for ResNet-18 and the 1st & 2nd layers for ResNet-50) produce sparser activation maps than the output layer (i.e., 2nd layer for ResNet-18 and 3rd layer for ResNet-50).

In addition, all residual blocks are divided into four stages that have different output feature map sizes. For ResNet-50, the four stages are composed of blocks 0 - 2, 3 - 6, 7 - 12, and 13 - 15. Figure B.4b

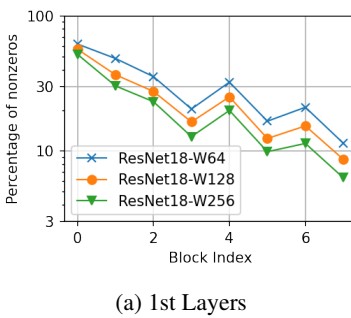
(a) 1st Layers

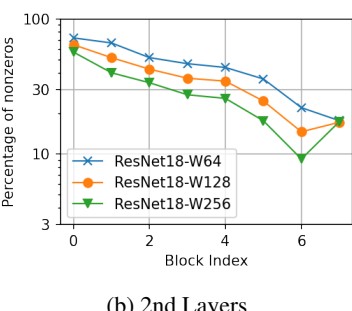
(b) 2nd Layers

Figure B.6: Effect of network width $\in \{64, 128, 256\}$ on sparsity level across layers of ResNet-18.

shows that there are patterns on how sparsity level varies within each stage and across the boundary of the stages.

- For the 1st layers, percentage of nonzeros decreases within each stage, and jumps up from the last layer of each stage to the first layer of next stage.
- For the 2nd layers, percentage of nonzeros decreases quickly at the beginning of each stage then becomes stable.
- For the 3rd layers, percentage of nonzeros tend to increase slightly within each stage, and jumps down from the last layer of each stage to the first layer of next stage.

Such observations may help to understand the role of each stage in ResNets.

Comparing the percentage of nonzero entries in ResNets (shown in Figure B.4a and Figure B.4b) and for Transformers (shown in Figure 2b), both of which are trained on ImageNet-1k, we see that ResNets produce much denser activation maps with more than 10% nonzero entries in all layers. One possible explanation is that ResNet uses batch normalization (BN) before each activation function, while Transformer's MLP does not have BN before the activation function. To understand the effect of BN on sparsity, we conduct an experiment with BN in ResNet removed. Because ResNet cannot be effectively trained without BN, we decrease the learning rate from standard ResNet training by a factor of 10. Moreover, we add a learnable scalar multiplier that is initialized as 0 to all the residual branches, following the study in Bachlechner et al. (2021); Qi et al. (2020). The results for comparing with standard ResNet are reported in Figure B.5, where to separate the effect of using a smaller learning rate, we also compare with the method of training a regular ResNet but with a small learning rate compared to standard training. The two subfigures of Figure B.5 show the effect of width on sparsity of the first and second layers in each residual block, respectively. It can be observed that, removing BN does not significantly change the sparsity level, except for small set of layers.

Meanwhile, the trend that larger models are sparser for Transformers (see Section 2.2) holds for ResNets as well, as seen in Figure B.6. Here, we vary the width of ResNet-18 by multiplying the number of output channels of each convolutional layer by a factor of 1 (for width = 64), 2 (for width = 128), and 4 (for width = 256). The two subfigures show the effect of width on sparsity of the first and second layers in each residual block, respectively. In both cases, wider models have smaller percentage of nonzero entries across all layers, except for the very last layer (i.e., the 2nd layer in block #7 shown in Figure B.6b).

**Sparsity and Residual Learning.** We provide a study on the effect of residual connections on activation sparsity. Each Transformer block contains two types of residual connections: the one that is in parallel with the attention blocks, and the one that is in parallel with the MLP blocks. We focus on the residual connection parallel to the MLP blocks. We perform two different studies.

- *Effect of shortcut connection.* Towards that, we train two T5-Large models, one using the vanilla Transformer block and the other with residual connection removed for the Transformer block on `encoder layer 6` (i.e., the 7th encoder layer, as we count from 0). There is a 1.6% evaluation accuracy drop with the latter model compared to the former model.

  The percentage of nonzero entries of these two Transformers are presented in Figure B.7 for the encoder layers and in Figure B.8 for the decoder layers. It can be seen that in `encoder layer 6` for which the residual connection is removed, the sparsity has a very different trend during

training compared to the corresponding layer of the vanilla Transformer. Moreover, the sparsity level at all other layers also changes, though to a much smaller extend.

- *Effect of initialization scale of the residual branch.* Many works have found that having the residual branch initialized at a smaller scale helps with stabilizing and accelerating the training of residual (convolutional) networks (Goyal et al., 2017; Zhang et al., 2019) and Transformers (Touvron et al., 2021). Here for simplicity we consider the idea from Bachlechner et al. (2021); Qi et al. (2020) where a trainable scalar multiplier that is initialized at zero is applied to the residual branch (a.k.a., ReZero).

We consider ViT trained on ImageNet-21k with ReZero added to the MLP modules. We find that this increases the training accuracy from 46.15% to 46.85% but reduces the validation accuracy from 47.58% to 46.75%. We plot the sparsity level of ViT with ReZero and compare it with the vanilla ViT in Figure B.9. It can be seen that ReZero reduces the percentage of nonzeros in layers near the network output.

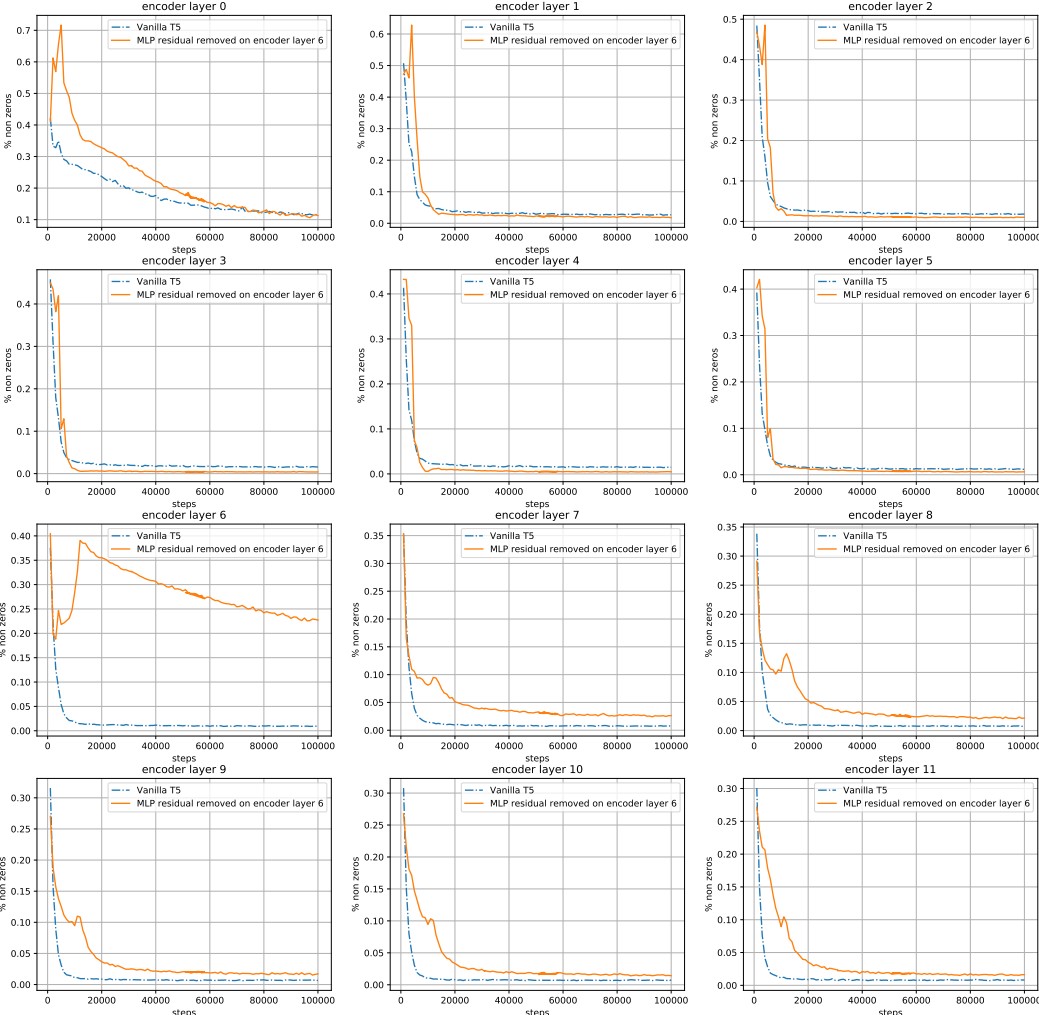

Figure B.7: Percentage of nonzero entries in activation maps in vanilla T5-Large and in a T5-Large with the residual connection parallel to MLP removed in the 7th encoder layer (i.e., `encoder layer 6`). Different subplots correspond to different *encoder* layers (see Figure B.8 for results on *decoder* layers). The `encoder layer 6`, which has its residual connection removed, shows a significant difference in both sparsity and the trend of sparsity during training. Sparsity level in other layers changes from vanilla T5-Large as well, though to a smaller extent.

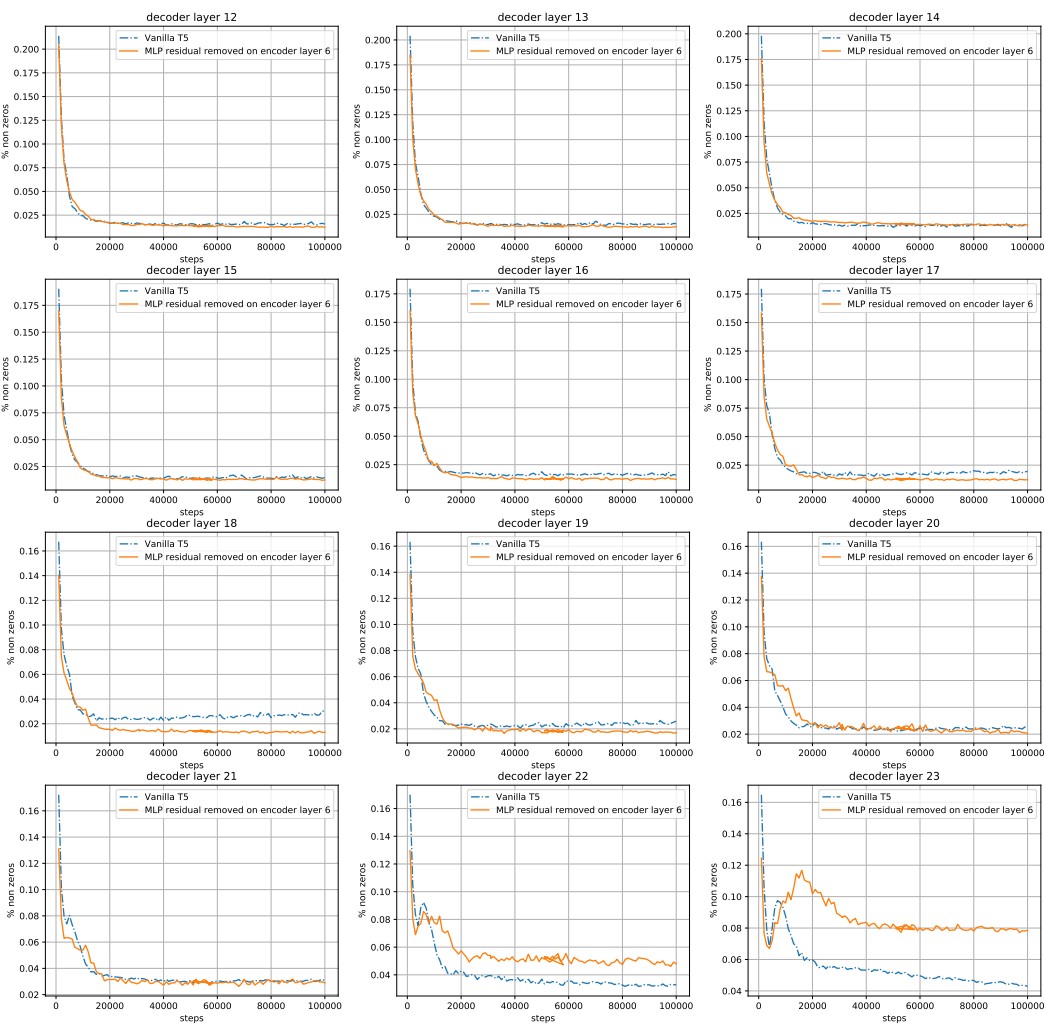

Figure B.8: Same setup as Figure B.7, but showing the results for the last 12 layers of the decoder.

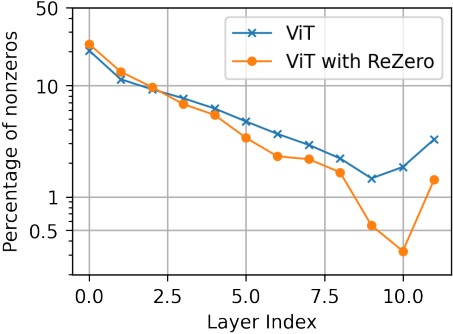

Figure B.9: Effect of initialization scale of the residual branch. We add a scalar multiplier that is initialized at 0 on the residual branch (a.k.a., ReZero Bachlechner et al. (2021)) of the MLP modules of a ViT, and train the model on ImageNet-21k. The percentage of nonzero entries is compared with those obtained with a regular ViT.

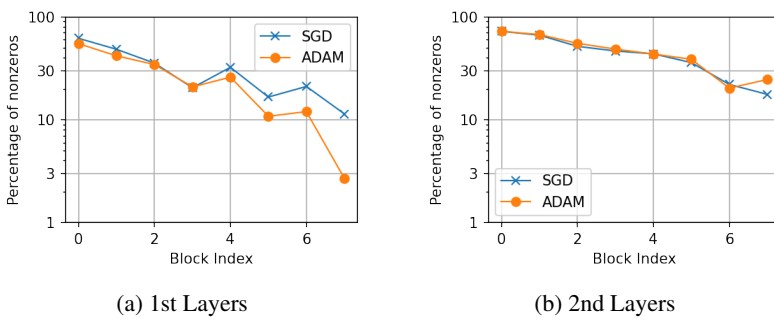

Figure B.10: Effect of optimizer on sparsity level across layers of ResNet-18.

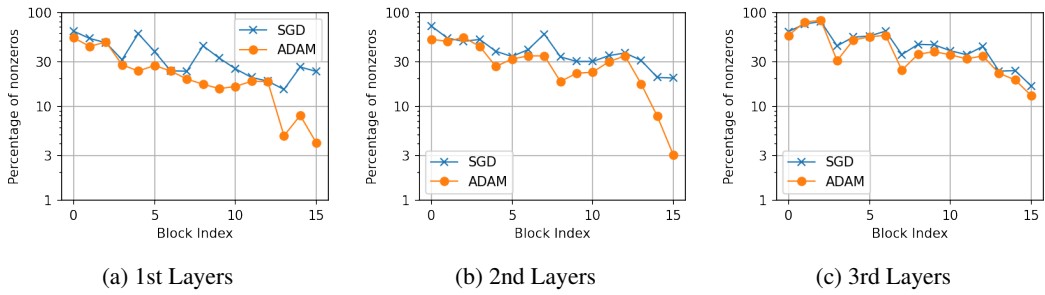

Figure B.11: Effect of optimizer on sparsity level across layers of ResNet-50.

## B.2 SPARSITY AND OPTIMIZER

Transformers are usually trained using ADAM or its variants as the optimizer (Kingma & Ba, 2015). It may be curious to ask whether the emergence of sparsity is specific to such optimizers and whether other optimizers, such as stochastic gradient descent (SGD), also leads to sparse activation maps. However, we find that SGD cannot effectively train Transformer architectures such as T5 and ViT. Hence, we study the effect of optimizer on activation sparsity by looking at ResNet trained on ImageNet-1k following the setup in Section B.1, since both SGD and ADAM can effectively train the network. To train ResNet with ADAM, we use the same hyper-parameters as those used in SGD, with the only difference being that the optimizer is ADAM with $\beta_1 = 0.9, \beta_2 = 0.999$. To make the comparison with SGD fair, we tune the base learning rate for ADAM and select $3e-3$, which is the one that gives the highest training accuracy among the set of $\{1e-4, 3e-4, 1e-3, 3e-3, 1e-2\}$. The training accuracy obtained by ADAM with base learning rate $3e-3$ is similar to that obtained by SGD, namely, 67.8% by ADAM vs 69.3% by SGD with ResNet-18, and 75.0% by ADAM vs 78.5% by SGD with ResNet-50.

The results for ResNet-18 and ResNet-50 are presented in Figure B.10 and Figure B.11, respectively. For ResNet-18, we see that ADAM leads to a smaller percentage of nonzero entries particularly towards the output of the network for the first layers of each residual block. In contrast, ADAM and SGD have very similar sparsity level at the second layers of each residual block. Similar observation holds for ResNet-50, where the percentage of nonzero entries is smaller with ADAM for the first and second layers of each residual block, while for the third layer the sparsity level does not change much.

## B.3 SPARSITY IN FINETUNING

In this section we show that activation sparsity not only occurs after model pretraining but persists after further finetuning on downstream tasks. Here we take a T5 that has been pretrained on C4 as described in Section 1.3, and finetune the model on a open domain Natural Question (Kwiatkowski et al., 2019) QA task. We follow the set up in Li et al. (2022), where the retrieved passages are independently encoded by the encoder, and then passed to the decoder via cross attention. The decoder takes the question as the prefix and produces the answer. The decoder is a standard auto-

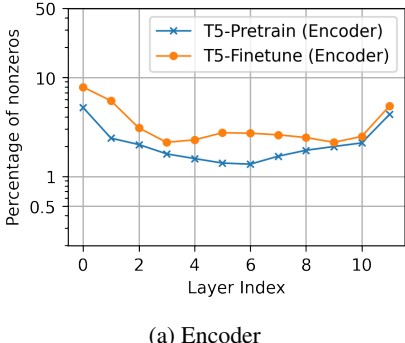

(a) Encoder

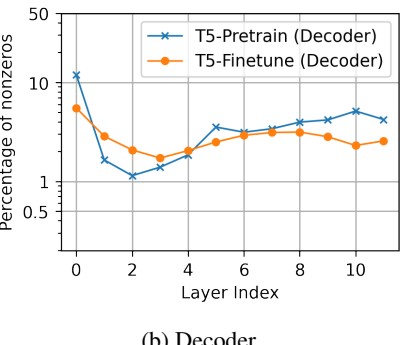

(b) Decoder

Figure B.12: Percentage of nonzero entries across different layers of trained T5 after pretraining vs after finetuning on a question answering task. Left: Results on encoder layers. Right: Results on decoder layers.

regressive setup, despite passing the question as a prefix. We used the same Wikipedia passages retrieved by FiD (Izacard & Grave, 2020) which uses DPR (Karpukhin et al., 2020) as retriever. For this experiment, we used 20 passages for each question. When calculating the sparsity level, we ignored the activation produced from paddings, for both encoder and decoder. The sparsity level of different encoder and decoder layers after finetuning is reported in Figure B.12 and is compared to those before the finetuning. It can be seen that finetuning does not drastically change the sparsity level.

## C  ADDITIONAL RESULTS ON BENEFITS OF SPARSITY

### C.1  TOP-$k$ DOES NOT SIGNIFICANTLY AFFECT TRAINING CONVERGENCE

One may argue that the convergence of Top-$k$ Transformer can be slower than that of a vanilla Transformer because, with fewer neurons being activated, the amount of parameters that have nonzero gradient associated with each training sample is smaller. Here we plot the training curves for ViT as well as Top-$k$ ViT with $k \in \{64, 128, 256\}$ on ImageNet-21K (this is the same setup as the experiments in Figure 5b), and report the results in Figure C.1. It can be observed that taking Top-$k$ does not significantly reduce convergence speed, particularly when $k$ is relatively large (e.g., $k = 256$).

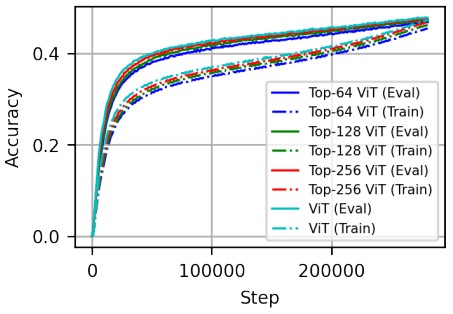

Figure C.1: Learning curves for results reported in Figure 5b. ViT and Top-$k$ ViT have similar convergence rate.

### C.2  BENEFITS OF SPARSITY PERSISTS WITH $\ell_1$-NORM INDUCED SPARSITY

While Top-$k$ thresholding is used in Section 3.3 to demonstrate the benefit of sparsity, we show that other means of obtaining sparsity, such as an explicit $\ell_1$ norm regularization, also provides such benefits.

We experiment with ViT for ImageNet-1k classification under the same setup as in Section 3.3. Here instead of the Top-$k$ ViT, we train a regular ViT but with an additional loss term, which is the sum of the $\ell_1$ norm of all activation maps of ViT across all layers. We refer to the method as L1-ViT. We vary the weight $\lambda$ on the $\ell_1$ loss in the set $\lambda \in \{0.1, 0.5, 1.0\}$ to control the strength of the regularization, and denote the corresponding methods as L1-ViT-$\{0.1, 0.5, 1.0\}$.

The sparsity level, natural accuracy, robust accuracy under input perturbation, and ECE of L1-ViT are reported in Table C.1. We see that with $\lambda = 0.1$ or $0.5$, the averaged percentage of nonzero entries do not change much, but already demonstrates performance gain in terms of accuracy under input perturbation and calibration without hurting the natural accuracy. Using a $\lambda = 1.0$ drastically reduces

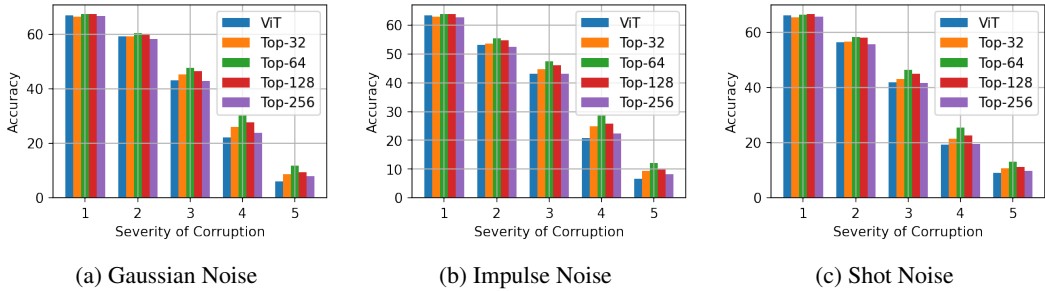

(a) Gaussian Noise  (b) Impulse Noise  (c) Shot Noise

Figure C.3: Performance of Top-$k$ ViT on corrupted ImageNet-1k test data with Gaussian noise (left), impulse noise (middle), and shot noise (right), each under five severity levels. Top-$k$ improves robustness for all noise types and on all corruption levels with a suitable choice of $k$.

Table C.1: Evaluation of ViT with a varying weight $\in \{0.1, 0.5, 1.0\}$ on a $\ell_1$ regularization upon activation maps for ImageNet-1k classification in terms of 1) averaged percentage of nonzero entries in activation maps across all layers, 2) natural accuracy (i.e., on ImageNet-1k evaluation set), 3) robust accuracy under input perturbation with additive {Gaussian, Impulse, Shot} noise, and 4) calibration error measured by ECE.

| Methods | Avg. Perc. of Nonzeros | Natural Accuracy | Accuracy under Input Perturbation | | | Expected Calibration Error (ECE) |
|---|---|---|---|---|---|---|
| | | | Gaussian | Impulse | Shot | |
| ViT | 5.67% | 74.85% | 39.54% | 37.37% | 38.56% | 8.42% |
| L1-ViT-0.1 | 5.66% | 75.02% | 40.70% | 38.52% | 39.56% | 8.37% |
| L1-ViT-0.5 | 5.76% | 74.83% | 42.46% | 40.64% | 40.96% | 8.22% |
| L1-ViT-1.0 | 1.60% | 73.21% | 40.26% | 38.01% | 38.95% | 6.34% |

the percentage of nonzero entries but it offers a better performance in terms of accuracy under input perturbation and calibration compared to ViT.

## C.3 Additional Results for Calibration

Under the same setup as in Section 3.3, we report the calibration of Top-$k$ ViT during training and with varying values of $k$, and report the results in Figure C.2. At the beginning of training the model has a low ECE because the output probabilities are mostly uniformly distributed across all classes hence the model is not confident, and that its prediction is purely random hence wrong with high probability. The model tends to become overly confident as the training progresses, hence the ECE increases particularly towards the end of training. What we can observe is that Top-$k$ enables the Transformer to be more calibrated when compared to a vanilla Transformer, particularly for small values of k. The results with $k = 128$ and its comparison with the vanilla ViT is also presented in Table 1.

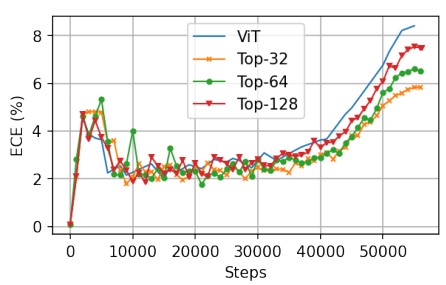

Figure C.2: Confidence calibration of Top-$k$ ViT for ImageNet-1k classification.

## C.4 Additional Results for Robustness to Input Perturbation

Under the same setup as in Section 3.3, we report the performance of Top-$k$ ViT with varying value of $k$ in Figure C.3. We can see that Top-$k$ ViT offers a performance gain over the vanilla ViT, particular with $k = 64$ or $k = 128$ and the severity level is high. The averaged performance over all severity levels of each corruption type is reported in Table 1.

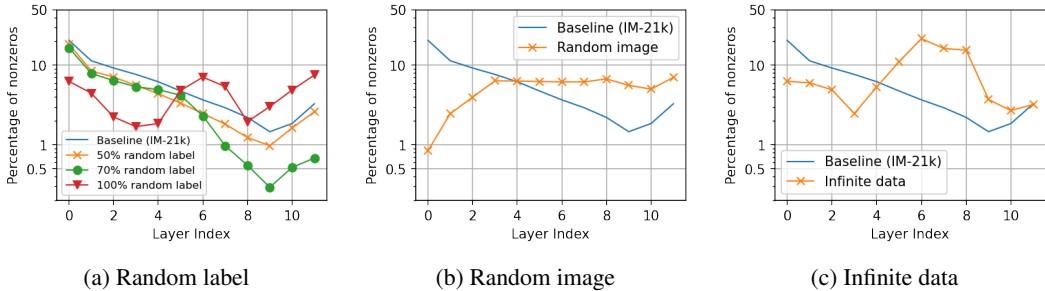

|  (a) Random label  |  (b) Random image  |  (c) Infinite data  |

Figure D.1: Percentage of nonzero entries in ViT trained on ImageNet-21k (IM-21K) with (a) *random labels* where $p\%$ labels are replaced by labels drawn from a uniform distribution with $p \in \{50\%, 70\%, 100\%\}$, (b) *random images* where each image is replaced by one where the pixels are drawn from i.i.d. uniform distribution in $[-1, 1]$, and (c) *infinite data* where sufficient training data is generated by drawing random image and random label pairs so that the model is never trained on the same pair twice.

## D    SPARSITY FROM TRAINING DYNAMIC?

In this section we study the causes of sparsity in activation maps of trained Transformers. Towards that, in Section D.1, D.2, and D.3 , we present a set of hypotheses and design corresponding experiments to validate or disprove them. We discuss the observation from the experiments and draw a tentative conclusion in Section D.4 on attributing sparsity to the training dynamic, with theoretical evidence.

### D.1    SPARSITY FROM LABELS?

Transformers are usually trained via supervised learning (e.g., using the ImageNet dataset for ViT) or self-supervised learning (e.g., using the span corruption task for T5). In both cases, the training labels provide a pertinent and meaningful description of the corresponding training data (e.g., image for ViT and text for T5). Sparsity may arise because the label set provides a structured organization of the massive training data, hence the training dataset admits a compact representation. This motivates us to formulate the following hypothesis.

**Hypothesis D.1** (Sparsity from labels). *Sparsity in trained Transformers arises from the labels for Transformer training, e.g., human annotations in supervised training or generated labels from data itself in self-supervised learning.*

We use a *random label* experiment with ViT for image classification to test Hypothesis D.1. Specifically, we generate a new training dataset by replacing $p\%$ of the labels in the ImageNet-21k dataset with random labels drawn uniformly at random from the set of all possible labels, where $p$ is varied to examine the effects. With such a dataset, the labels for a certain percentage of images do not provide a meaningful description for the content of the image. Hence, if Hypothesis D.1 is valid, then the activation map will become dense.

The sparsity level of ViT trained on the random label datasets is shown in Figure D.1a. It can be seen that the percentage of activated neurons decreases with an increasing percentage of label noise up to $70\%$. An even higher label noise level at $100\%$ changes the sparsity level across layers as the shallow layers (i.e., layers 0 - 4) becomes sparser, while the deep layers (i.e., layers 5 - 11) becomes denser. Nonetheless, even with $100\%$ label noise, all layers have $< 10\%$ activated neurons.

### D.2    SPARSITY FROM DATA?

While modern image and text data are often of high-dimensional, their intrinsic degree of freedom is much smaller, i.e., they are low-dimensional and admit compact representations (Vidal et al., 2015; Wright & Ma, 2022). Hence, even if the labels do not provide meaningful descriptions of the data, it may still be possible that Transformers extract low-dimensional structures from data and produce compact representations in the form of sparse activation maps. This motivates the following hypothesis.

**Hypothesis D.2** (Sparsity from natural data). *Sparsity in trained Transformers arises from natural training data (e.g., images for ViT and texts for T5).*

We use a *random image* experiment to test Hypothesis D.2. With the ImageNet-21k dataset, we replace each image with a random image generated by drawing pixel values from an i.i.d. Uniform distribution in the range of [0, 255], and use these images (instead of the original images in ImageNet-21k) for model training. Such random images do not contain any low-dimensional structures nor compact representations.

The percentage of nonzero entries of a ViT trained on random image dataset is shown in Figure D.1b. It can be seen that the first four layers become sparser while the last few layers become relatively denser compared to training with natural images in ImageNet-21k. Nonetheless, all layers have $< 10\%$ activated neurons.

## D.3    SPARSITY FROM DATA-FITTING?

Modern deep neural networks are often *over-parameterized*, with sufficient capacity to fit practical training datasets and obtain close-to-zero training error. There is evidence suggesting that this result holds true even if the data and label are generated in random (Zhang et al., 2021). Hence, there is the possibility that sparsity arises because the training data, even if generated in random, is scarce relative to the scale of modern over-paremeterized models.

**Hypothesis D.3** (Sparsity from data-fitting). *Sparsity in trained Transformers arises from the fact that models have more than sufficient capacity to fit training data of practical scale.*

To test Hypothesis D.3, we design an *infinite data* experiment where the amount of training data is infinitely large so that any practical Transformer becomes under-parameterized relative to the data and cannot fit the data. The way we generate infinite training data is to sample images with random pixels as in the random image experiment, and for each image we sample a random label as in the random label experiment. Moreover, we generate sufficient amount of such training data to make sure that the model never sees the same data point twice during the training. The number of training iterations in the infinite data experiment is kept the same as that of the random image and random label experiments.

The result of this experiment is presented in Figure D.1c. It can be seen that the first four layers produce sparser activation maps, while middle layers with index 4 - 7 are considerably denser compared to the baseline with near 10% to 20% nonzero entries.

## D.4    DISCUSSION: SPARSITY FROM TRAINING DYNAMIC?

The results of random label, random image, and infinite data experiments in Figure D.1 show that labels, data, and data-fitting as conjectured in Hypothesis D.1, D.2, and D.3, respectively, all affect the sparsity level of the activation map. Nonetheless, none of them fully explains the emergence of sparsity since for all results in Figure D.1, the percentage of nonzero entries is considerably smaller than at the initialization (i.e., 50%).

Our results point to the possibility that sparsity comes from the training dynamic. Namely, at early training stage with any training data and a random initialization of network parameters, the descending direction of the gradient on the Transformer parameters tends to point to a regime where their MLPs produce sparse activation maps. In the following, we provide theoretical evidence for this argument by looking at the gradient on the positive activation maps for a DNN with last two layers being a ReLU followed by a fully connected layer. In particular, we have the follow result.

**Theorem D.1.** *Let $f(\boldsymbol{x}; \boldsymbol{V}, \boldsymbol{\theta}) : \mathbb{R}^n \to \mathbb{R}^K$ be a neural network given by*

$$f(\boldsymbol{x}) = \boldsymbol{V}\sigma\big(\boldsymbol{p}(\boldsymbol{x}; \boldsymbol{\theta})\big), \tag{D.1}$$

*where $\boldsymbol{V} = [\boldsymbol{v}_1, \ldots, \boldsymbol{v}_{d_{f\!f}}] \in \mathbb{R}^{K \times d_{f\!f}}$ is network parameter for the last layer drawn from a random distribution, $\sigma()$ is the ReLU activation function, and $\boldsymbol{p}(\boldsymbol{x}; \boldsymbol{\theta})$ denotes all other layers with parameter $\boldsymbol{\theta}$. We write $\boldsymbol{p} = \boldsymbol{p}(\boldsymbol{x}; \boldsymbol{\theta})$ for simplicity.*

- *Consider the mean squared error (MSE) loss $\ell_{MSE}(f(\boldsymbol{x}), \boldsymbol{y}) \doteq \frac{1}{2}\|f(\boldsymbol{x}) - \boldsymbol{y}\|_2^2$, where $\boldsymbol{y}$ is an arbitrary vector independent of $\boldsymbol{V}$. Assume that $\boldsymbol{V}$ satisfies*

$$\mathbb{E}\left[\boldsymbol{V}\right] = \boldsymbol{0}, \quad and \quad \mathbb{E}\left[\langle \boldsymbol{v}_i, \boldsymbol{v}_j \rangle\right] \begin{cases} = 0, & if \ i \neq j, \\ > 0, & otherwise^6. \end{cases} \tag{D.2}$$

  *If there exist an $i^*$ such that $p_{i^*} > 0$, then we have*

$$\mathbb{E}\left[\frac{\partial \ell_{MSE}(f(\boldsymbol{x}), \boldsymbol{y})}{\partial \boldsymbol{p}_{i^*}}\right] > 0, \tag{D.3}$$

  *where the expectation is taken with respect to randomness in $\boldsymbol{V}$.*

- *Consider the cross-entropy (CE) loss $\ell_{CE}(f(\boldsymbol{x}), \boldsymbol{y}) = -\langle \boldsymbol{y}, \log \frac{\exp(f(\boldsymbol{x}))}{\langle \exp(f(\boldsymbol{x})), I \rangle} \rangle$, where $\boldsymbol{y}$ is an arbitrary vector that sums up to one and independent of $\boldsymbol{V}$. Assume that the entries of $\boldsymbol{V}$ are drawn from independent distributions, the probability of any entry of $\boldsymbol{V}$ being 0 is less than 1, and $\mathbb{E}\left[\boldsymbol{V}\right] = \boldsymbol{0}$. If there exist an $i^*$ such that $p_{i^*} > 0$, then we have*

$$\mathbb{E}\left[\frac{\partial \ell_{CE}(f(\boldsymbol{x}), \boldsymbol{y})}{\partial \boldsymbol{p}_{i^*}}\right] > 0, \tag{D.4}$$

  *where the expectation is taken with respect to randomness in $\boldsymbol{V}$.*

The proof of Theorem D.1 is provided in Appendix E. Theorem D.1 states that the gradient of either the MSE or CE loss with respect to any positive activation $p_{i^*}$ is positive in expectation. Hence, any training algorithm based on negative gradient directions tends to reduce the magnitude of such positive activations, which will lead to a smaller training loss. Here, the expectation is taken with respect to the randomness in the last layer parameter $\boldsymbol{V}$. Hence, our result can be considered as an analysis for DNNs at initialization where weights are often chosen randomly from a fixed distribution. In particular, the required properties for the distribution of $\boldsymbol{V}$ in Theorem D.1 for both MSE and CE losses are satisfied by commonly used initialization methods, such as the one in He et al. (2015). On the other hand, Theorem D.1 does not apply to subsequent training iterations since the label $\boldsymbol{y}$ is no longer independent of $\boldsymbol{V}$. However, it can be seen empirically from Figure 1 that the trend of a decreasing percentage of nonzero entries persists for a certain number of iterations during the beginning of training until such a percentage reaches a low level and stays relatively stable until the end of training.

# E  PROOF OF THEOREM D.1

*Proof of Theorem D.1.* For an arbitrary loss $\ell(f(\boldsymbol{x}), \boldsymbol{y})$, we have

$$\frac{\partial \ell}{\partial p_{i^*}} = \left\langle \frac{\partial \ell}{\partial f}, \frac{\partial f}{\partial p_{i^*}} \right\rangle = \left\langle \frac{\partial \ell}{\partial f}, \boldsymbol{v}_{i^*} \right\rangle. \tag{E.1}$$

First, Consider $\ell = \ell_{MSE}$. We have

$$\frac{\partial \ell_{MSE}}{\partial f} = f(\boldsymbol{x}) - \boldsymbol{y} = \sum_i \sigma(p_i) \cdot \boldsymbol{v}_i - \boldsymbol{y}. \tag{E.2}$$

Plugging this into (E.1), we obtain

$$\begin{aligned}
\frac{\partial \ell_{MSE}}{\partial p_{i^*}} &= \left(\sum_i \sigma(p_i)\langle \boldsymbol{v}_i, \boldsymbol{v}_{i^*} \rangle\right) - \langle \boldsymbol{v}_{i^*}, \boldsymbol{y} \rangle \\
&= \left(\sum_{i \neq i^*} \sigma(p_i)\langle \boldsymbol{v}_i, \boldsymbol{v}_{i^*} \rangle\right) + \sigma(p_{i^*})\langle \boldsymbol{v}_{i^*}, \boldsymbol{v}_{i^*} \rangle - \langle \boldsymbol{v}_{i^*}, \boldsymbol{y} \rangle
\end{aligned} \tag{E.3}$$

Taking the expectation, and noting the conditions in (D.2), we have

$$\mathbb{E}\left[\frac{\partial \ell_{MSE}}{\partial p_{i^*}}\right] = 0 + \sigma(p_{i^*})\mathbb{E}\left[\langle \boldsymbol{v}_{i^*}, \boldsymbol{v}_{i^*} \rangle\right] + 0 > 0. \tag{E.4}$$

This finishes the proof for MSE loss.

---

$^6$This requirement is generally satisfied unless the probability of $\boldsymbol{v}_i = \boldsymbol{0}$ is 1.

In the rest of the proof we consider $\ell = \ell_{CE}$. We have

$$\frac{\partial \ell_{CE}}{\partial f} = \frac{\exp(f(\boldsymbol{x}))}{\langle \exp(f(\boldsymbol{x})), \mathbf{1} \rangle} - \boldsymbol{y} = \frac{\exp(\sum_i \sigma(p_i) \cdot \boldsymbol{v}_i)}{\langle \exp(\sum_i \sigma(p_i) \cdot \boldsymbol{v}_i), \mathbf{1} \rangle} - \boldsymbol{y}. \tag{E.5}$$

Plugging this into (E.1), we obtain

$$\frac{\partial \ell_{CE}}{\partial p_{i^*}} = \frac{\langle \exp(\sum_i \sigma(p_i) \cdot \boldsymbol{v}_i), \boldsymbol{v}_{i^*} \rangle}{\langle \exp(\sum_i \sigma(p_i) \cdot \boldsymbol{v}_i), \mathbf{1} \rangle} - \langle \boldsymbol{v}_{i^*}, \boldsymbol{y} \rangle \tag{E.6}$$

For the enumerator in the first term on the RHS of the equation above, we have

$$\left\langle \exp\left(\sum_i \sigma(p_i) \cdot \boldsymbol{v}_i\right), \boldsymbol{v}_{i^*} \right\rangle = \sum_m \left( v_{i^*,m} \cdot \exp\left(\sum_i \sigma(p_i) \cdot v_{im}\right) \right)$$

$$= \sum_m \left( v_{i^*,m} \cdot \exp\left(p_{i^*} \cdot v_{i^*,m}\right) \cdot \exp\left(\sum_{i \neq i^*} \sigma(p_i) \cdot v_{i,m}\right) \right) \tag{E.7}$$

Plugging this into (E.6) and denoting

$$C_m^{(1)} = \exp\left(\sum_{i \neq i^*} \sigma(p_i) \cdot v_{i,m}\right),$$

we obtain

$$\frac{\partial \ell_{CE}}{\partial p_{i^*}} = \sum_m \left( \frac{v_{i^*,m} \cdot \exp\left(p_{i^*} \cdot v_{i^*,m}\right) \cdot C_m^{(1)}}{\langle \exp(\sum_i \sigma(p_i) \cdot \boldsymbol{v}_i), \mathbf{1} \rangle} \right) - \langle \boldsymbol{v}_{i^*}, \boldsymbol{y} \rangle \tag{E.8}$$

For the denominator in the first term on the RHS of the equation above, we have

$$\left\langle \exp\left(\sum_i \sigma(p_i) \cdot \boldsymbol{v}_i\right), \mathbf{1} \right\rangle = \sum_{m'} \exp\left(\sum_i \sigma(p_i) \cdot v_{im'}\right)$$

$$= \sum_{m'} \left( \exp\left(p_{i^*} \cdot v_{i^*,m'}\right) \cdot \exp\left(\sum_{i \neq i^*} \sigma(p_i) \cdot v_{im'}\right) \right)$$

$$= \exp\left(p_{i^*} \cdot v_{i^*,m}\right) \cdot \exp\left(\sum_{i \neq i^*} \sigma(p_i) \cdot v_{i,m}\right)$$

$$+ \sum_{m' \neq m} \left( \exp\left(p_{i^*} \cdot v_{i^*,m'}\right) \cdot \exp\left(\sum_{i \neq i^*} \sigma(p_i) \cdot v_{im'}\right) \right) \tag{E.9}$$

Plugging this into (E.8) and denoting

$$C_m^{(2)} = \exp\left(\sum_{i \neq i^*} \sigma(p_i) \cdot v_{i,m}\right), \tag{E.10}$$

$$C_m^{(3)} = \sum_{m' \neq m} \left( \exp\left(p_{i^*} \cdot v_{i^*,m'}\right) \cdot \exp\left(\sum_{i \neq i^*} \sigma(p_i) \cdot v_{im'}\right) \right), \tag{E.11}$$

we obtain

$$\frac{\partial \ell_{CE}}{\partial p_{i^*}} = \sum_m \left( \frac{v_{i^*,m} \cdot \exp\left(p_{i^*} \cdot v_{i^*,m}\right) \cdot C_m^{(1)}}{\exp\left(p_{i^*} \cdot v_{i^*,m}\right) \cdot C_m^{(2)} + C_m^{(3)}} \right) - \langle \boldsymbol{v}_{i^*}, \boldsymbol{y} \rangle. \tag{E.12}$$

Taking expectation with respect to $\boldsymbol{V}$ on both sides, and using the assumption that all entries of $V$ are independent, we have

$$
\mathbb{E}\left[\frac{\partial \ell_{CE}}{\partial p_{i^*}}\right] = \sum_m \mathbb{E}\left[\frac{v_{i^*,m} \cdot \exp\left(p_{i^*} \cdot v_{i^*,m}\right) \cdot C_m^{(1)}}{\exp\left(p_{i^*} \cdot v_{i^*,m}\right) \cdot C_m^{(2)} + C_m^{(3)}}\right] - \mathbb{E}\left[\langle \boldsymbol{v}_{i^*}, \boldsymbol{y}\rangle\right]
$$

$$
= \sum_m \mathbb{E}_{\{v_{i,l}|(i,l)\neq(i^*,m)\}}\left[\mathbb{E}_{v_{i^*,m}}\left[\frac{v_{i^*,m} \cdot \exp\left(p_{i^*} \cdot v_{i^*,m}\right) \cdot C_m^{(1)}}{\exp\left(p_{i^*} \cdot v_{i^*,m}\right) \cdot C_m^{(2)} + C_m^{(3)}}\right]\right] - \mathbb{E}\left[\langle \boldsymbol{v}_{i^*}, \boldsymbol{y}\rangle\right].
$$
(E.13)

In above, $\mathbb{E}_{v_{i^*,m}}\,[]$ means expectation with respect to $v_{i^*,m}$, and $\mathbb{E}_{\{v_{i,l}|(i,l)\neq(i^*,m)\}}\,[]$ means expectation with respect to all other entries in $\boldsymbol{V}$. Note that $C_m^{(1)}, C_m^{(2)}$, and $C_m^{(3)}$ are independent of $v_{i^*,m}$. By Lemma E.1 and using the assumption that the expectation of $\boldsymbol{V}$ is zero, we have

$$
\mathbb{E}_{v_{i^*,m}}\left[\frac{v_{i^*,m} \cdot \exp\left(p_{i^*} \cdot v_{i^*,m}\right) \cdot C_m^{(1)}}{\exp\left(p_{i^*} \cdot v_{i^*,m}\right) \cdot C_m^{(2)} + C_m^{(3)}}\right] > 0,
$$
(E.14)

and

$$
\mathbb{E}\left[\langle \boldsymbol{v}_{i^*}, \boldsymbol{y}\rangle\right] = 0.
$$
(E.15)

Plugging the above two relations into (E.13), we obtain

$$
\mathbb{E}\left[\frac{\partial \ell_{CE}}{\partial p_{i^*}}\right] > 0.
$$
(E.16)

$\square$

The following lemma is used in the proof above.

**Lemma E.1.** *Let* $\mathsf{V}$ *be a random variable with a probabilistic density function* $p(v)$ *that satisfies* $P(\mathsf{V} = 0) \neq 1$. *Let* $C_1, C_2, C_3$ *and* $p$ *be positive numbers. Then,*

$$
\mathbb{E}\left[\frac{C_1 \mathsf{V} \cdot \exp(pv)}{C_2 \exp(p\mathsf{V}) + C_3}\right] > \frac{C_1}{C_2 + C_3}\mathbb{E}\left[\mathsf{V}\right].
$$
(E.17)

*Proof.* We may calculate the expectation by using the probabilistic density function $p(v)$ as

$$
\mathbb{E}\left[\frac{C_1 \mathsf{V} \cdot \exp(p\mathsf{V})}{C_2 \exp(p\mathsf{V}) + C_3}\right] = \mathbb{E}\left[\frac{C_1 \mathsf{V}}{C_2 + C_3 \exp(-p\mathsf{V})}\right] = \int_{-\infty}^{\infty}\frac{C_1 v}{C_2 + C_3 \exp(-pv)}p(v)dv \doteq \int_{-\infty}^{\infty} g(v) \cdot vp(v)dv.
$$
(E.18)

Since $g(v)$ is monotonically increasing for $v \in \mathbb{R}$, we have $g(v) \geq g(0)$ for $v \geq 0$ and $g(v) \leq g(0)$ for $v \leq 0$. Hence,

$$
\int_{-\infty}^{0} g(v) \cdot vp(v)dv \geq g(0)\int_{-\infty}^{0} vp(v)dv,
$$
(E.19)

$$
\int_{0}^{\infty} g(v) \cdot vp(v)dv \geq g(0)\int_{0}^{\infty} vp(v)dv.
$$
(E.20)

Moreover, since $P(\mathsf{V} = 0) \neq 1$, there exists an interval $(a, b)$ such that $\int_a^b p(v)dv > 0$. Without loss of generality we assume that $b > a \geq 0$. Then,

$$
\int_{a}^{b} g(v) \cdot vp(v)dv > g(0)\int_{a}^{b} vp(v)dv.
$$
(E.21)

That is, the inequality in (E.20) holds with strict inequality. Hence we have

$$
\mathbb{E}\left[\frac{C_1 \mathsf{V} \cdot \exp(p\mathsf{V})}{C_2 \exp(p\mathsf{V}) + C_3}\right] = \int_{-\infty}^{\infty} g(v) \cdot vp(v)dv > g(0)\mathbb{E}\left[\mathsf{V}\right] = \frac{C_1}{C_2 + C_3}\mathbb{E}\left[\mathsf{V}\right].
$$
(E.22)

$\square$

# F    INSIGHTS FROM SPARSITY IN MLPS

We study the sparsity of activation maps in two-layer MLPs. By showing that sparsity emerges, the result here extends the scope of prevalence of activation sparsity from modern DNNs to two-layer

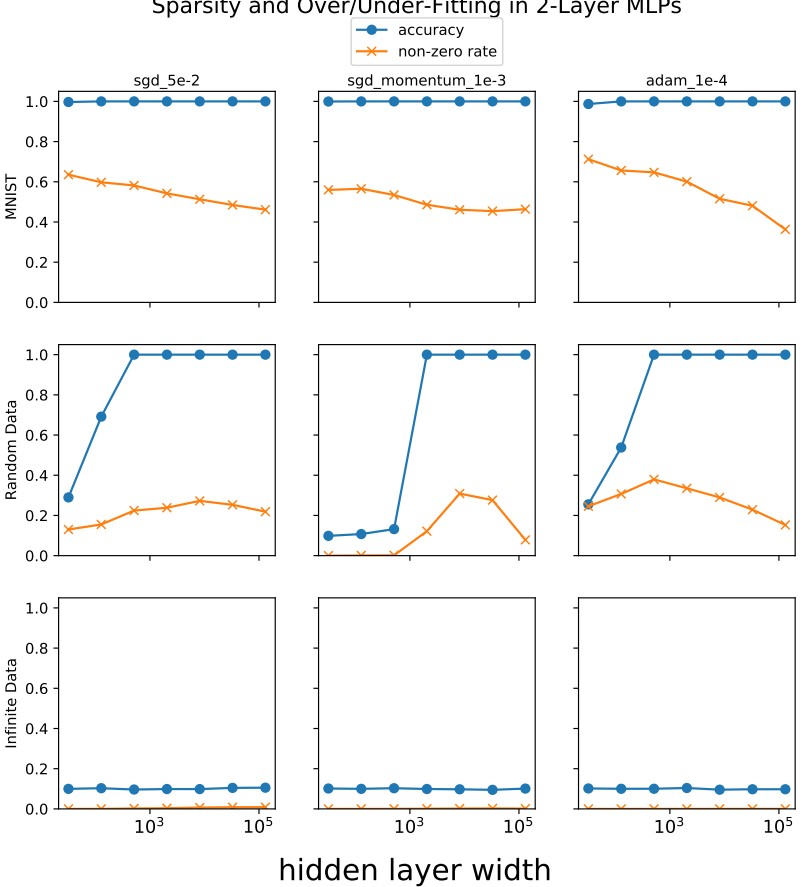

Figure F.1: Training accuracy and percentage of nonzero entries (both on the y-axis) in activation maps of two-layer MLPs of varying width (on the x-axis, in log scale) after 200 epochs of training. Rows correspond to different training datasets, and columns correspond to different training algorithms.

MLPs which are one of the simplest neural network architectures. Moreover, by training such two-layer MLPs with different types of data, we provide additional insights on the causes for emergence of sparsity.

**Datasets.** We conduct our experiment with the MNIST dataset, which contains 60,000 grey scale images of handwritten digits. Similar to the experiment in Section D, we also consider a dataset with *random* data, as well as a dataset with *infinite* data. For the random data, we replace each image of MNIST with a random one drawn from sampling i.i.d. pixels from uniform distribution, and each label with a random class amongst 10. Note that the image-label pairs are fixed throughout training. For the infinite data, the random images and random labels are generated on-the-fly, representing a random dataset of infinite size.

**Models and Training.** We train two-layer MLPs with ReLU activation maps with varying width (i.e., hidden dimension): 32, 128, 512, 2048, 8192, 32768 and 131072. We use three different optimizers: SGD, SGD with momentum, and Adam, all for 200 epochs (for the infinite data case, we use the same number of iterations as that for training on MNIST and random data). We find that 200 epochs is sufficient for the reported metrics to converge in most of the cases.

**Results.** We report training accuracy and the percentage of nonzero entries in the intermediate activation map (i.e., non-zero rate) at the end of the training in Figure F.1. We have the following observations.

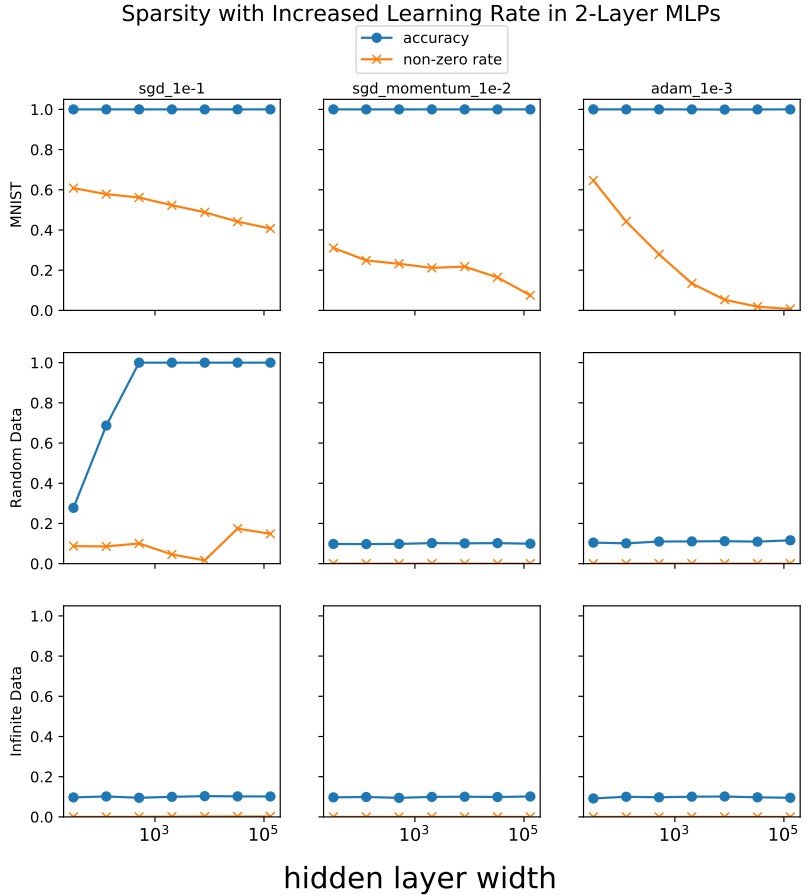

Figure F.2: The same as in F.1 except that for each optimizer we use a larger learning rate.

- For random data, we observe a *uni-modal* shaped curve for sparsity level. Namely, when the model width is small hence the model cannot well-fit the training data, the percentage of nonzero entries is small. As the model width increases, where the model is able to fit the training data evidenced by the fact that the training accuracy increases, we observe that the percentage of nonzero entries starts to increase. However, as we further increase the model size in the regime where model is able to perfectly fit the training data, we see that the percentage of nonzeros starts to decrease.

- For infinite data, where the model cannot fit the training data (hence training accuracy is 0.1 which is the same as result from random guessing), the percentage of nonzero entries is close to 0. This is aligned with the result of random data experiment with a small model width.

- For MNIST, where the model of varying width in our experiment is able to fit the training data, we observe that the percentage of nonzero entries decreases. This trend aligns with the random data experiment with large model width

The evidence above suggest that the sparsity level may be associated with the under- and over-parameterization of the models. Namely, the percentage of nonzero entries is the highest when the model size is close to the point that the model can start to fit the training data (i.e., the interpolation threshold), and is lower in both under and over-parameterized regimes. It may be intriguing to note that a similar pattern exists for the variance (as in the bias-variance tradeoff) curve of deep learning models, which as shown in Yang et al. (2020) to exhibit a uni-modal shape as well. Such a connection may help us understand the interplay between generalization and sparsity of activation in deep learning models.

Finally, in Figure F.2 we report the results obtained with optimizers with larger learning rates compared to those used in Figure F.1. It can be observed that larger learning rate produces sparser activation maps.

