# OpenReview forum: "The Lazy Neuron Phenomenon: On Emergence of Activation Sparsity in Transformers"
_ICLR.cc/2023/Conference — ICLR 2023 poster_

### Official Review · Reviewer_HpEq · 2022-10-24

**Confidence:** 3
**Correctness:** 3
**Technical Novelty And Significance:** 3
**Empirical Novelty And Significance:** 3
**Recommendation:** 8

**Clarity, Quality, Novelty And Reproducibility:**

This paper's motivation is based on statistical observation. The analysis is insightful and well supported by some experiments.
As T5 and ViT are public available models, this paper is reproducible.


**Details Of Ethics Concerns:**

No ethics concerns.

**Strength And Weaknesses:**

Strength:
(1)This paper observed the sparsity in Transformers and this sparsity is emergent without any explicit design.

(2) This paper found that sparsity in Transformers is a prevalent phenomenon, and can improve efficiency, robustness and calibration.

(3) This paper further presented Top-k Transformer, a simple modification to Transformer architecture that allows control on the sparsity level for all model inputs throughout training.

Weakness:
(1) This paper only shows sparsity on the classification task. Is the sparsity existing for downstream tasks? such as detection and segmentation.

(2) Tab 1 shows that Top-128 ViT is on par with ViT for natural accuracy while is significantly better for model robustness and calibration. How about the other Top-k results? (please move some experiments from the appendix).


**Summary Of The Paper:**

This work demonstrates the natural emergence of sparsity in commonly used Transformer models.

This paper proposed  Top-k thresholding to enforce sparsity, which brings robustness of training with erroneous annotations, less sensitivity to input noise/perturbation, and better confidence calibration of the predictions.

**Summary Of The Review:**

This paper observed the sparsity in Transformers and proposed Top-k thresholding to enforce sparsity. Based on the proposed method, the efficiency, robustness and calibration are improved. However, this paper only considered the classification task, and I'd like to see the sparsity existing in the downstream tasks.

---

> ### Author Response · Authors · 2022-11-16
> **Response to Reviewer HpEq**
>
> > This paper only shows sparsity on the classification task. Is the sparsity existing for downstream tasks? such as detection and segmentation.
>
> Thanks for the suggestion. We performed an experiment to examine the effect of finetuning on the emergence of sparsity. In particular, we take a T5 pretrained on C4 and finetune it for a question answering task. We find that finetuning hardly changes the sparsity level. We added this result to Appendix B.3. For detection and segmentation, ViT does not have a default setup for such finetuning tasks, so we’d like to postpone such a study to after the rebuttal period.
>
> > Tab 1 shows that Top-128 ViT is on par with ViT for natural accuracy while is significantly better for model robustness and calibration. How about the other Top-k results? (please move some experiments from the appendix).
>
> The effect of k for robustness and calibration is provided in Fig. C.2 and Fig. C.1. They show that 1) Top-64 offers the highest robust accuracy, followed by Top-128 and Top-32, followed by Top-256 which is roughly on par with vanilla Transformer; 2) reducing k from 128 to 32 monotonically increases calibration. We added a short description of such results in Sec. 3, but due to space limitations we unfortunately cannot move the details to the main paper. Meanwhile, we note that robustness and calibration are only part of our contribution. Another major part of the paper was dedicated to discussing the ubiquitousness of sparse activation and on its implication for efficiency.

---

> > ### Comment · Reviewer_HpEq · 2022-12-11
> > **Response to the Authors**
> >
> > Thanks for the authors' responses, which address my former two concerns. I keep my former rating of acceptance.

---

### Official Review · Reviewer_x5F1 · 2022-10-26

**Confidence:** 3
**Correctness:** 3
**Technical Novelty And Significance:** 2
**Empirical Novelty And Significance:** 2
**Recommendation:** 5

**Clarity, Quality, Novelty And Reproducibility:**

The authors propose interesting insights into the sparsity patterns of  transformers. However, the contributions of the paper seems to be limited to only proposing to take the top k values in hidden layers of MLPs. In particular, from Fig. 6 we see that this does not necessarily lead to running time performance. Moreover, the in Fig. 5 we see that the baseline networks slightly outperform the proposed method in most cases.

**Strength And Weaknesses:**

Strengths:

(a) The work presents an interesting application that can be easily applied to all transformer based models.
(b) The paper is in general well-written and easy to follow.
(c) The authors conduct additional experiments on the robustness of the models.

Weaknesses:

(a) From Fg. 6 I see that in quite a few cases using this "sparse" model does not lead to improvement in inference time. For instance for T5 Large all the numbers are negative. This undermines the main goal of this model, which was to improve running time.
(b) On page 6, the authors discuss the possibility of using approximate algorithms for finding the top K values with sublinear running times in hidden dimension. However, it is not clear to me of the authors actually used this approximate method or not.
(c) Minor comments/questions:
    (c1) Are the results in Fig. 1 for train set or test set?
    (c2) Is dim in the legend of Fig. 4 the same as dff?
    (c3) What is dff in Fig. 5?
    (c4) On page 9, change "many work..." to "many works..."



**Summary Of The Paper:**

The authors study the role of sparsity in deep neural networks that are based on transformers. In particular, the authors show that for a trained transformer only a small percentage of the hidden neurons in Multi-Layer Perceptrons (MLP) are non-zero. Motivated by that they propose a variant of the transformer networks, for a MLP layer only the top k hidden neurons are kept. They conduct numerical experiments for two instances of transformer networks for images and one for natural language processing. When comparing their version with the base models, they show that this model achieves comparable metrics, when the top 128 neurons (k=128) are kept. They further investigate the robustness of this model by introducing noisy labels and adding noise to the inputs. Overall, this new model appears to be outperform the baseline in this noisy conditions.

**Summary Of The Review:**

Overall, the authors show intersting insights into the transformers sparsity patterns. However, I believe that the the current numerical results are not convincing to immediately adopt this variant.

---

> ### Author Response · Authors · 2022-11-16
> **Response to Reviewer x5F1**
>
> > From Fig. 6, in quite a few cases using this "sparse" model does not lead to improvement in inference time… This undermines the main goal of this model, which was to improve running time.
>
> First, we want to clarify that the main goal of this paper is not restricted to reducing running time. Specifically, our goal is to show that activation sparsity is an important notion for Transformers as 1) it occurs very broadly (Sec. 2), 2) can be used to improve efficiency (Sec. 3), and 3) can be used to improve robustness and calibration (Sec. 4).
>
> Second, in terms of efficiency, the wall-time reduction result reported in the paper serves only as a proof-of-concept but does not fully utilize the benefit of sparsity. The reason is that our method requires computation with unstructured sparsity and data-dependent sparsity patterns, which are usually not well supported on computation hardware such as TPUs and GPUs. Please also see our general response “Clarification on significance of wall-time reduction and the main contribution of the paper”.
>
> > It is not clear to me of the authors actually used this approximate method (i.e. nearest neighbor search) or not.
>
> We discuss the approximate nearest neighbor search as a potential future research direction to further improve efficiency of MLPs with sparse activations. We do not use it in the experiments in this paper. We clarified this at the end of section 3.2.
>
> > Are the results in Fig. 1 for train set or test set?
>
> These are on the training set. We added a clarification in Sec. 1.1 and Sec. 1.3.
>
> > Is dim in the legend of Fig. 4 the same as dff?
>
> Yes. We have updated the figure to include more informative legend. Thanks!
>
> > What is dff in Fig. 5?
>
> We added a Table A.1 in the appendix that summarizes the configuration of all models for the reader’s convenience.
>
> > On page 9, change "many work..." to "many works..."
>
> We have corrected it. Thanks!

---

### Official Review · Reviewer_KPJJ · 2022-10-26

**Confidence:** 3
**Correctness:** 3
**Technical Novelty And Significance:** 3
**Empirical Novelty And Significance:** 4
**Recommendation:** 8

**Clarity, Quality, Novelty And Reproducibility:**

This work has high quality and high clarity. This work has originality in analyzing the sparsity of transformer models based on my knowledge. The results should be reproducible because the authors provide amounts of experiments.

**Strength And Weaknesses:**

Strength:

(1)  The authors show the natural emergence of sparse activation in commonly used Transformer models.

(2)  Inspired by the experiments on sparsity, a top-k transformer is proposed to obtain good performance and generalization.

(3)  Extensive experiments are conducted to support their arguments.


Weaknesses:

This work shows the emergence of sparse activation in Transformer models. I have some questions as follows.

(1)  In Sec. 1.3, the authors state that they use ReLU as the activation function instead of GeLU.  However, what is the performance gap between ReLU and GeLU? If GeLU is employed, Does sparsity still exist? Furthermore, the authors also propose a top-k transformer. The authors should list the specific performance with the top-k transformer and the original GeLU for comparisons.

(2)  The authors choose the top k value for the transformer. However, it might affect the data parallelism thereby reducing the inference speed. The authors should provide a reasonable comparison, e.g., throughput.


**Summary Of The Paper:**

This work shows that sparsity is prevalent in transformer models, including T5 model for NLP and ViT model for vision with extensive experiments. Such a discovery might be suggesting that the law of parsimony is playing a role in Transformers even though they are not explicitly designed so. Furthermore, a top-k transformer is proposed by using top-k thresholding, which achieves good generalization.

**Summary Of The Review:**

To sum up, the work shows the emergence of sparse activation in transformer models with amounts of experiments and proposes a top-k transformer based on this discovery, which is promising. Therefore, I tend to vote accept. Some experiments for reasonable comparisons should also be conducted.


-------- After rebuttal --------

I have read the authors' responses and other reviewers' comments. I keep my 'accept' rating.

---

> ### Author Response · Authors · 2022-11-16
> **Response to Reviewer KPJJ**
>
> > What is the performance gap between ReLU and GeLU?
>
> We evaluated Top-1 accuracy on the evaluation dataset of ImageNet-21K and found that the gap is very small, namely, 47.78% with GeLU vs 47.58% with ReLU. We added this clarification to footnote 3.
>
> > If GeLU is employed, does sparsity still exist?
>
> We added a comparison of preactivation histograms for BERT Base models with different activation functions GeLU, Sigmoid and Tanh (in Appendix Figure B.3). We observe a similar distribution of preactivation values as ReLU with GeLU and Sigmoid activations. However Tanh activation has a different distribution of preactivation values. With Tanh activation, the network doesn't show sparsity and the accuracy is significantly worse in comparison to ReLU/GeLU.
>
> > The authors should list the specific performance with the top-k transformer and the original GeLU for comparisons.
>
> We note that Top-K Transformer is agnostic to the choice of activation function, i.e., Top-K can be added to the Transformer with either ReLU or GeLU activation functions. In Fig. 5, we showed that adding Top-K to a Transformer with ReLU does not hurt performance with an appropriate K. Hence, we expect that adding Top-K to a Transformer with GeLU does not hurt the performance with an appropriate K either. We want to omit such a comparison for now since the experiment takes a long time to run, but we'd be happy to include results in the next version.
>
> > The authors choose the top k value for the transformer. However, it might affect the data parallelism thereby reducing the inference speed. The authors should provide a reasonable comparison, e.g., throughput.
>
> The Top-K Transformer drastically reduces FLOP count, providing huge headroom for improving efficiency. In practice, as the reviewer mentions, many factors such as data parallelism may affect the throughput or wall-time improvement. However, hardware co-evolves with algorithms. It is far beyond the scope of this paper to design hardware. We hope that our study serves as a good motivation for designing future hardware that takes full advantage of the intrinsic sparsity of neural networks.

---

> > ### Comment · Reviewer_KPJJ · 2022-11-29
> > **Response to the Authors**
> >
> > Thanks for the authors' responses, which address my concerns. It is better that the authors can provide the quantitative inference speed evaluation.

---

### Official Review · Reviewer_cu8i · 2022-10-26

**Confidence:** 2
**Correctness:** 4
**Technical Novelty And Significance:** 2
**Empirical Novelty And Significance:** 2
**Recommendation:** 5

**Clarity, Quality, Novelty And Reproducibility:**

The paper is clear, but I'm not sold on the novelty. I don't know what the takeaway of this paper should be. The authors have run many experiments but it's hard to decipher a core message.


**Strength And Weaknesses:**

+ Reducing the number of operations in a dense matrix multiplication makes sense as a route to speedups.
+ The writing is easy to understand.
+ Exhaustive empirical evaluation

- I'm not convinced by reading the paper that sparsity it the solution to more computationally efficient models. Right above section 3.3, the authors claim a 10% wall-time reduction. Is this significant? I wonder if it's worth retraining a model to be sparse for only a 10% wall-time reduction.
- I'm curious why the authors mainly studied the activations of the MLP portion of transformers and not so much the activation maps prior to the MLP.
- Section 3.3 contains interesting information but feels rushed and too short. I was hoping the authors could expand on how much sparsity improves robustness to noise and confidence calibration.


**Summary Of The Paper:**

The authors study sparsity in the MLP portion of transformers as a means to speed up inference time computation. They also show that sparsity enables robustness to input noise.


**Summary Of The Review:**

I am on the fence about accepting this paper. It contains a lot of interesting experiments but is lacking a strong message. I hope the authors can address some of the weaknesses: I am open to increasing my score.

---

> ### Author Response · Authors · 2022-11-16
> **Response to Reviewer cu8i**
>
> > The authors claim a 10% wall-time reduction. Is this significant?
>
> The wall-time reduction is small because commonly used hardware for deep learning is not optimized for sparse computation.  However, hardware co-evolves with algorithms. In the future, it is possible that new sparsity aware hardware can take full advantage of the sparse computation in our Top-K Transformers. Please also see our general response “Clarification on significance of wall-time reduction and the main contribution of the paper”.
>
> > I'm curious why the authors mainly studied the activations of the MLP portion of transformers and not so much the activation maps prior to the MLP.
>
> Sparsity in other activation maps is definitely a problem of great interest and worthy of an independent investigation by itself. We plan to study it in the future. However, we do feel that the results on the MLP hidden layer are of special interest because they are potentially the widest layer in a transformer and consume a significant amount of FLOPs while performing relatively simple computation.
>
> >  the authors could expand on how much sparsity improves robustness to noise and confidence calibration.
>
>  We have a more detailed study on how sparsity improves robustness and confidence calibration but due to space limit we can only put them into appendix C.
>
> > I don't know what the takeaway of this paper should be. The authors have run many experiments but it's hard to decipher a core message.
>
> The takeaway message is: Activation sparsity is an important phenomenon for Transformers as 1) it occurs in most scenarios  (Sec. 2), and 2) can be used to improve efficiency (Sec. 3), robustness, and calibration (Sec. 4). Please also see our general response “Clarification on significance of wall-time reduction and the main contribution of the paper”.

---

### Official Review · Reviewer_urTS · 2022-10-31

**Confidence:** 4
**Correctness:** 4
**Technical Novelty And Significance:** 2
**Empirical Novelty And Significance:** 3
**Recommendation:** 6

**Clarity, Quality, Novelty And Reproducibility:**

The paper is well written. Appendix includes all details to reproduce the results.

**Strength And Weaknesses:**

Strengths:
- Many experiments emphasize that this phenomenon happens for a variety of architectures and tasks.
- The appendix offers some very good insights. The relevance of the optimizer and the residual connections are the first candidates that fall in mind when first reading the paper and could be emphasized/briefly mentioned in the main text as well.
- It is nice to see that depending on the task and the rank of the representation required, different sparsity patterns with layer index can be observed.

Weaknesses:
- As mentioned, time-wall performance is not guaranteed apart from unbatched greedy decoding for larger models.
- Would be beneficial to discuss more (dis)similarities with sparsity in the attention matrix. I suspect similar results to Table 1 would come out if the Top-k operator was applied to the attention matrix.

Questions/Remarks:
- Would be interesting to see what happens when residual connections weights are learned, as e.g. in ReZero [1].
- The Role of the optimizer is discussed extensively in the Appendix. [2] might offer an alternative to train Transformers via SGD, which would be a valuable comparison.
- It is not clearly described what the legend in Figure 5 means. I guess (Train) indicates that the model was trained from scratch with a fixed k? If Top-k Transformer models are trained from scratch, it would be interesting to know if more steps are required to reach convergence.
- Studying the relationship with ResNets is not fair, as ResNets have batch normalization layers before the activations. Comparison with MLP should also include residual connections to be fairer.

[1]: Bachlechner, Thomas, et al. "Rezero is all you need: Fast convergence at large depth." Uncertainty in Artificial Intelligence. PMLR, 2021.
[2]: Noci, Lorenzo, et al. "Signal Propagation in Transformers: Theoretical Perspectives and the Role of Rank Collapse." arXiv preprint arXiv:2206.03126 (2022).

**Summary Of The Paper:**

The paper experimentally discusses the emergence of sparse activations in the feed-forward layer of Transformer neural networks. The authors highlight that sparse activations emerge for a variety of tasks and architectures. They then discuss how sparsity can be taken advantage of to reduce FLOP count and the implications of sparsity for robustness under different sources of noise.

**Summary Of The Review:**

All in all the paper offers some valuable insights on the activation sparsity in (Transformer) networks. Plethora of experiments demonstrate that this phenomenon is present across architectures and tasks. Claims regarded FLOP efficiency are briefly mentioned, but not adequately backed. A new Top-k layer is proposed, with limited insights given regarding drops in expected accuracy and expected gains in wall time inference. Robustness claims are also linked to previous work. The authors do make attempts to explain possible causes (mainly in the appendix). Contributions are incremental and not totally clear.

---

> ### Author Response · Authors · 2022-11-16
> **Response to Reviewer urTS**
>
> > The relevance of the optimizer and the residual connections … could be emphasized/briefly mentioned in the main text as well.
>
> Thanks for the suggestion. We added a sentence for it in Sec. 1.2.
>
> > Would be beneficial to discuss more (dis)similarities with sparsity in the attention matrix.
>
> We completely agree with the reviewer and indeed sparsity in the attention matrix has been well studied, showing that Top-K is usually harmless in attention computation (see e.g. [a, b]). For a given query, the softmax in attention computation results in small values for most keys. These observations have driven a lot of research into designing efficient Transformers that reduce attention computation cost [c]. Our hope is the findings in this paper about sparsity in the MLP activations will drive similar efforts towards better understanding and reducing computational cost for the MLP layers in Transformers.
>
> [a] Zhao, Guangxiang, et al. "Explicit sparse transformer: Concentrated attention through explicit selection." arXiv preprint arXiv:1912.11637 (2019).
> [b] Gupta, Ankit, et al. "Memory-efficient Transformers via Top-$ k $ Attention." arXiv preprint arXiv:2106.06899 (2021).
> [c] Tay, Yi, et al. "Efficient transformers: A survey." ACM Computing Surveys (CSUR) (2020).
>
> > time-wall performance is not guaranteed apart from unbatched greedy decoding for larger models.
>
> This is because commonly used hardware for deep learning is not optimized for sparse computation. However, hardware co-evolves with algorithms. In the future, it is possible that new sparsity aware hardware can take full advantage of the sparse computation in our Top-K Transformers. Please also see our general response “Clarification on significance of wall-time reduction and the main contribution of the paper”.
>
> > Would be interesting to see what happens when residual connections weights are learned, as e.g. in ReZero [1].
>
> We conducted this experiment and added the results in Fig. B.9. It shows that ReZero helps to decrease the number of nonzero entries for layers near the network output.
>
> > The Role of the optimizer is discussed extensively in the Appendix. [2] might offer an alternative to train Transformers via SGD, which would be a valuable comparison.
>
> We thank the reviewer for bringing to our attention this interesting work on training Transformers with SGD. We plan to explore using it for training T5 / ViT and compare with the results of Adam reported in our paper, but due to the time limit of the rebuttal period, we leave such an exploration to after the rebuttal period,
>
> > It is not clearly described what the legend in Figure 5 means.
>
> For Top-K T5, (Train) and (Eval) means the training and evaluation accuracy for a T5 model trained with fixed k from the beginning of training. Top-K T5 and regular T5 are trained using the same number of iterations. We have added a plot of the learning curves in Fig. C.1, to show that Top-K does not significantly affect the convergence speed in model training.
>
> > Studying the relationship with ResNets is not fair, as ResNets have batch normalization layers before the activations. Comparison with MLP should also include residual connections to be fairer.
>
> We thank the reviewer for this interesting comment. To isolate the effect of batch normalization, we conducted an experiment for a ResNet without it and added the results in Fig. B.5. It can be seen that batch normalization does not significantly change the sparsity level.
>
> Regarding MLP: If the reviewer refers to the MLP-Mixer results in Fig. B.1, we note that residual connection is used in MLP-Mixer as well. If the reviewer refers to the MLP results in Sec. F, we would like to mention that those results are not meant to compare the sparsity level between MLP and Transformers. In particular, Transformers are usually trained on large scale data such as ImageNet and C4, but the MLP in Sec. F is trained on MNIST and randomly generated data, so comparing their sparsity level does not make much sense.

---

> > ### Comment · Reviewer_urTS · 2022-11-17
> > **Response to the authors**
> >
> > Thank you for the detailed feedback and the new experiments.
> >
> > All of my questions have been addressed.

---

### Author Response · Authors · 2022-11-16
**Clarification on significance of wall-time reduction and our main contribution**

We would like to thank all reviewers for taking the time to review our paper, and provide valuable feedback.

Reviewers urTS, cu8i, and x5F1 raised questions on the significance of our results for wall-time reduction. This also led to some doubts on our contribution (x5F1) and the main takeaway (cu8i) of the paper. We clarify on those points below.

First, our paper is far beyond a paper on introducing a new technique for improving model efficiency. The main message of our paper is that activation sparsity is an important notion for Transformers, because 1) it naturally occurs in most scenarios  (Sec. 2), 2) it can be used to improve efficiency (Sec. 3), and 3) it can be used to improve robustness and calibration (Sec. 4). We believe that such a study is of great value because sparsity (also known as Occam’s razor) is a fundamentally important concept for scientific discovery, and our findings may motivate fundamental improvement of deep learning by using sparsity in the future. We briefly discussed this point in Sec. 5.

Second, in terms of efficiency, the wall-time reduction result reported in the paper serves only as a proof-of-concept but does not fully utilize the benefit of sparsity. The reason is that our method requires computation with unstructured sparsity and data-dependent sparsity patterns, which are usually not well supported on computation hardware such as TPUs and GPUs. However, hardware co-evolves with algorithms. In the future, it is possible and it is our hope that our work motivates new sparsity aware hardware that can take full advantage of the sparse computation in our Top-K Transformers.

We have revised our writing in related sections (i.e., Sec. 1.2, beginning of Sec. 3, Sec. 3.2) to clarify on the points above.

---

### Author Response · Authors · 2022-12-06
**Further Discussions**

We would like to thank the reviewers again for their detailed and helpful reviews. We have provided responses to all the questions raised by the reviewers.

As the discussion period will end soon, we would like to check and see if there are any remaining issues or questions regarding our work. We would be happy to clarify points that are still unclear.

---

### Decision · Program_Chairs · 2023-01-20

**Decision:**

Accept: poster

**Justification For Why Not Higher Score:**

The novelty aspect of the contribution made in the paper is somewhat limited. This is mostly an empirical paper with some interesting results but it does not appear to be a breakthrough result that would justify an oral (or even a spotlight).

**Justification For Why Not Lower Score:**

The reviewers all agree this paper is worth publishing.

**Metareview: Summary, Strengths And Weaknesses:**

The paper experimentally discusses the emergence of sparse activations in the feed-forward layer of Transformer neural networks. This paper is mostly empirical and although it seems to have a limited degree of novelty, the reviewers do find some of the conclusions of the papers interesting.

The various concerns raised by the reviewers were mostly addressed by the authors so overall, this is an interesting contribution and I recommend acceptance. Note however that there was little discussion among reviewers following the author's response. The decision was therefore mostly taken based on my own reading of the paper and reviews, and the feedback that was given by the authors (which I found to be mostly satisfying).

One aspect that I find lacking in the paper is the discussion of prior work. Some reviewers (including Reviewer urTS) provided some additional references that are relevant to the paper and the authors should discuss them in their revised manuscript.

**Note From Pc:**

if the above contains the word "oral" or "spotlight" please see: "oral" presentation means -> notable-top-5% and "spotlight" means -> notable-top-25%. As stated in our emails, we are disassociating presentation type from AC recommendations

**Summary Of Ac-Reviewer Meeting:**

N/A